# Chemical and structural investigation of the paroxetine-human serotonin transporter complex

Jonathan A Coleman[1†], Vikas Navratna[1†], Daniele Antermite[2], Dongxue Yang[1], James A Bull[2], Eric Gouaux[1,3]*

[1]Vollum Institute, Oregon Health & Science University, Portland, United States; [2]Department of Chemistry, Imperial College London, Molecular Sciences Research Hub, London, United Kingdom; [3]Howard Hughes Medical Institute, Oregon Health & Science University, Portland, United States

**Abstract** Antidepressants target the serotonin transporter (SERT) by inhibiting serotonin reuptake. Structural and biochemical studies aiming to understand binding of small-molecules to conformationally dynamic transporters like SERT often require thermostabilizing mutations and antibodies to stabilize a specific conformation, leading to questions about relationships of these structures to the bonafide conformation and inhibitor binding poses of wild-type transporter. To address these concerns, we determined the structures of ΔN72/ΔC13 and ts2-inactive SERT bound to paroxetine analogues using single-particle cryo-EM and x-ray crystallography, respectively. We synthesized enantiopure analogues of paroxetine containing either bromine or iodine instead of fluorine. We exploited the anomalous scattering of bromine and iodine to define the pose of these inhibitors and investigated inhibitor binding to Asn177 mutants of ts2-active SERT. These studies provide mutually consistent insights into how paroxetine and its analogues bind to the central substrate-binding site of SERT, stabilize the outward-open conformation, and inhibit serotonin transport.

*For correspondence:
gouauxe@ohsu.edu

†These authors contributed equally to this work

**Competing interests:** The authors declare that no competing interests exist.

## Introduction

Serotonin or 5-hydroxytryptamine (5-HT) is a chemical messenger which acts on cells throughout the human body, beginning in early development and throughout adulthood (*Berger et al., 2009*). 5-HT acts as both a neurotransmitter and a hormone that regulates blood vessel constriction and intestinal motility (*Berger et al., 2009*). In the central nervous system, 5-HT is released from presynaptic neurons where it diffuses across the synaptic space and binds to 5-HT receptors, promoting downstream signaling and activating postsynaptic neurons (*Gether et al., 2006*; *Kristensen et al., 2011*). Thus, 5-HT is a master regulator of circuits, physiology and behavioral functions including the sleep/wake cycle, sexual interest, locomotion, thermoregulation, hunger, mood, and pain (*Berger et al., 2009*). 5-HT is cleared from synapses and taken into presynaptic neurons by the serotonin transporter (SERT), thus terminating serotonergic signaling (*Gether et al., 2006*; *Kristensen et al., 2011*; *Rudnick et al., 2014*). SERT resides in the plasma membrane of neurons and belongs to a family of neurotransmitter sodium symporters (NSSs) which also includes the dopamine (DAT) and norepinephrine transporters (NET) (*Gether et al., 2006*; *Kristensen et al., 2011*; *Rudnick et al., 2014*). NSSs are twelve transmembrane spanning secondary active transporters which utilize sodium and chloride gradients to energize the transport of neurotransmitter across the membrane (*Rudnick et al., 2014*; *Navratna and Gouaux, 2019*; *Yamashita et al., 2005*; *Figure 1a*).

The function of NSSs is modulated by a spectrum of small-molecule drugs, thus in turn controlling the availability of neurotransmitter at synapses. Selective serotonin reuptake inhibitors (SSRIs) are a

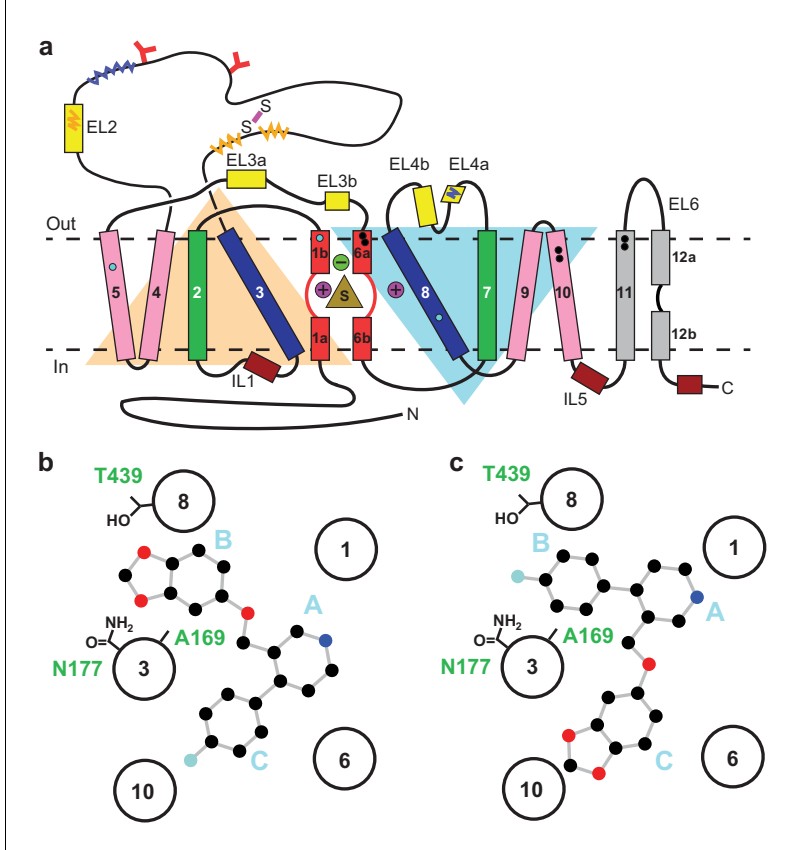

**Figure 1.** Topology of SERT. (**a**) The substrate is bound at the central site (sand, triangle), near two sodium ions (purple, spheres +) and a chloride ion (green, sphere -). The light orange and light blue triangles depict pseudo two-fold symmetric helical repeats comprised of TM1-5 and 6–10, respectively. The disulfide bond (purple line) and *N*-linked glycosylation (red 'Y' shapes) in extracellular loop 2, along with sites of thermostable mutations (Tyr110Ala, TM1a; Ile291Ala, TM5; Thr439Ser, TM8) are also shown (cyan-filled circles). Structural elements involved in binding allosteric ligands are depicted as black-filled circles. Epitopes for the 8B6 and 15B8 Fab binding sites are in squiggly dark-blue and orange lines, respectively. (**b**) Schematic of the ABC pose of paroxetine bound to the central binding site, derived from the previously determined x-ray structures (*Coleman and Gouaux, 2018*; *Coleman et al., 2016a*). The transmembrane helices are shown with circles and mutated residues in subsite B are in sticks. c, The ACB pose of paroxetine bound to the central binding site of SERT predicted by molecular dynamics simulations and mutagenesis (*Abramyan et al., 2019*; *Slack et al., 2019*).

class of drugs which inhibit SERT and are used to treat major depression and anxiety disorders (*Cipriani et al., 2018*). Using x-ray crystallography and cryo-EM, we have determined structures of thermostabilized variants of human SERT complexed with SSRIs, which together explain many of the common features and differences associated with SERT-SSRI interactions (*Coleman and Gouaux, 2018*; *Coleman et al., 2016a*). SSRIs are competitive inhibitors that bind with high-affinity and specificity to a central substrate-binding site in SERT, preventing 5-HT binding and arresting SERT in an outward-open conformation (*Gether et al., 2006*; *Kristensen et al., 2011*; *Coleman et al., 2016a*).

The central site in NSSs is composed of three subsites: A, B, and C (*Wang et al., 2013*; *Figure 1b*). In all NSS-ligand structures, the amine group of ligands resides in subsite A and interacts with a conserved Asp residue (Asp98 in SERT). The heterocyclic electronegative group of the ligand is positioned in subsite B (*Navratna and Gouaux, 2019*). For example, the alkoxyphenoxy groups of reboxetine and nisoxetine (*Penmatsa et al., 2015*) in *Drosophila* DAT (dDAT) structures, the halophenyl groups of cocaine analogs in dDAT and *S*-citalopram in SERT, and the catechol derivatives in DCP-dDAT and sertraline-SERT all occupy subsite B (*Coleman and Gouaux, 2018*; *Coleman et al., 2016a*; *Wang et al., 2015a*). In addition to the central binding site, the activity of SERT and NSSs can also be modulated by small-molecules which bind to an allosteric site located in an extracellular

vestibule, typically resulting in non-competitive inhibition of transport (*Coleman et al., 2016a*; *Zhong et al., 2009*; *Wennogle and Meyerson, 1982*; *Plenge and Mellerup, 1985*).

Paroxetine is an SSRI which exhibits the highest known binding affinity for the central site of SERT ($70.2 \pm 0.6$ pM) compared to any other currently prescribed antidepressants (*Cool et al., 1990*). Despite its high affinity, paroxetine is frequently associated with serious side effects including infertility, birth defects, cognitive impairment, sexual dysfunction, weight gain, suicidality, and cardiovascular issues (*Nevels et al., 2016*). As a result, the mechanism of paroxetine binding to SERT has been studied extensively in order to design drugs with higher-specificity and less adverse side-effects. Despite these efforts, however, the binding pose of paroxetine remains a subject of debate (*Coleman and Gouaux, 2018*; *Coleman et al., 2016a*; *Abramyan et al., 2019*; *Davis et al., 2016*; *Slack et al., 2019*).

Paroxetine is composed of a secondary amine which resides in a piperidine ring, which in turn is connected to benzodioxol and fluorophenyl groups (*Figure 1b*). X-ray structures of the SERT-paroxetine complex revealed that the piperidine ring binds to subsite A while the benzodioxol and fluorophenyl groups occupy subsite B and C in the central site, respectively (*Coleman and Gouaux, 2018*; *Coleman et al., 2016a*) (ABC pose, *Figure 1b*). However, recent mutagenesis, molecular dynamics, and binding studies with paroxetine analogues suggest that paroxetine might either occupy ABC pose as observed in the crystal structure, or an ACB pose where the benzodioxol and fluorophenyl groups occupy subsite C and B of the central site respectively (*Abramyan et al., 2019*; *Slack et al., 2019*; *Figure 1c*). Paroxetine is also thought to interact with the allosteric site of SERT, albeit with low-affinity (*Plenge and Mellerup, 1985*). We have, however, been unable to visualize paroxetine binding at the allosteric site using structural methods. Our x-ray maps, by contrast, resolve a density feature at the allosteric site which instead resembles a molecule of detergent (*Coleman et al., 2016a*).

To resolve the ambiguity of paroxetine binding poses at the central binding site, we turned to paroxetine derivatives whereby the 4-fluoro group is substituted with either a bromine or an iodine group. Using transport and binding assays, anomalous x-ray diffraction, and cryo-EM, we have examined the binding poses of these paroxetine analogs and their interactions at the central site. Our studies provide key insights into the recognition of high-affinity inhibitors by SERT and the rational design of new small-molecule therapeutics.

## Results

To provide a robust molecular basis for the interaction of paroxetine (**1**) with SERT, we devised synthetic routes for two derivatives of paroxetine where the 4-fluoro moiety is substituted with either bromo (Br-paroxetine, **2**) or iodo (I-paroxetine, **3**) groups (*Figure 2a,b*). We envisaged the use of a C–H functionalization strategy to access enantiopure hydroxymethyl intermediates **I**, from readily available *N*-Boc (*R*)-nipecotic acid **4** (*Figure 2b*, Appendix 1). Transition metal-catalyzed C–H functionalization can promote the reaction of unactivated C(sp$^3$)–H bonds with the aid of a directing group (*He et al., 2017*; *Rej et al., 2020*; *Antermite and Bull, 2019*; *O' Donovan et al., 2018*; *Maetani et al., 2017*; *Chapman et al., 2016*). Here, C–H functionalization enabled installation of the appropriate aryl group on the pre-existing piperidine ring (*Antermite et al., 2018*), providing an attractive and short route to vary this functionality with inherent control of enantiomeric excess. In contrast, common methods for (–)-paroxetine synthesis can require the aromatic substituent to be introduced before stereoselective steps or ring construction, reducing flexibility of the process (*Slack et al., 2019*; *Johnson et al., 2001*; *Hughes et al., 2003*; *Brandau et al., 2006*; *Krautwald et al., 2014*; *Wang et al., 2015b*; *Kubota et al., 2016*; *Amat et al., 2000*). Nevertheless, during the preparation of this work, the synthesis of Br-paroxetine was reported using an asymmetric conjugate addition and its binding to SERT has been extensively studied (*Slack et al., 2019*; *Brandau et al., 2006*).

Our synthesis commenced with the C–H arylation of piperidine (–)−**5** bearing Daugulis' aminoquinoline amide directing group (*Zaitsev et al., 2005*) at C(3). Adapting our reported method (*Antermite et al., 2018*), Pd-catalyzed C–H functionalization was achieved in moderate yields using 4-bromoiodobenzene or 1,4-diiodobenzene in excess to prevent bis-functionalization, with palladium acetate, $K_2CO_3$ and pivalic acid (*Figure 2c*). The *cis*-arylated derivatives (+)−**6a** and (+)−**6b** were obtained with > 98% *ee* and complete C(4) selectivity. Minor enantiopure *trans*-functionalized

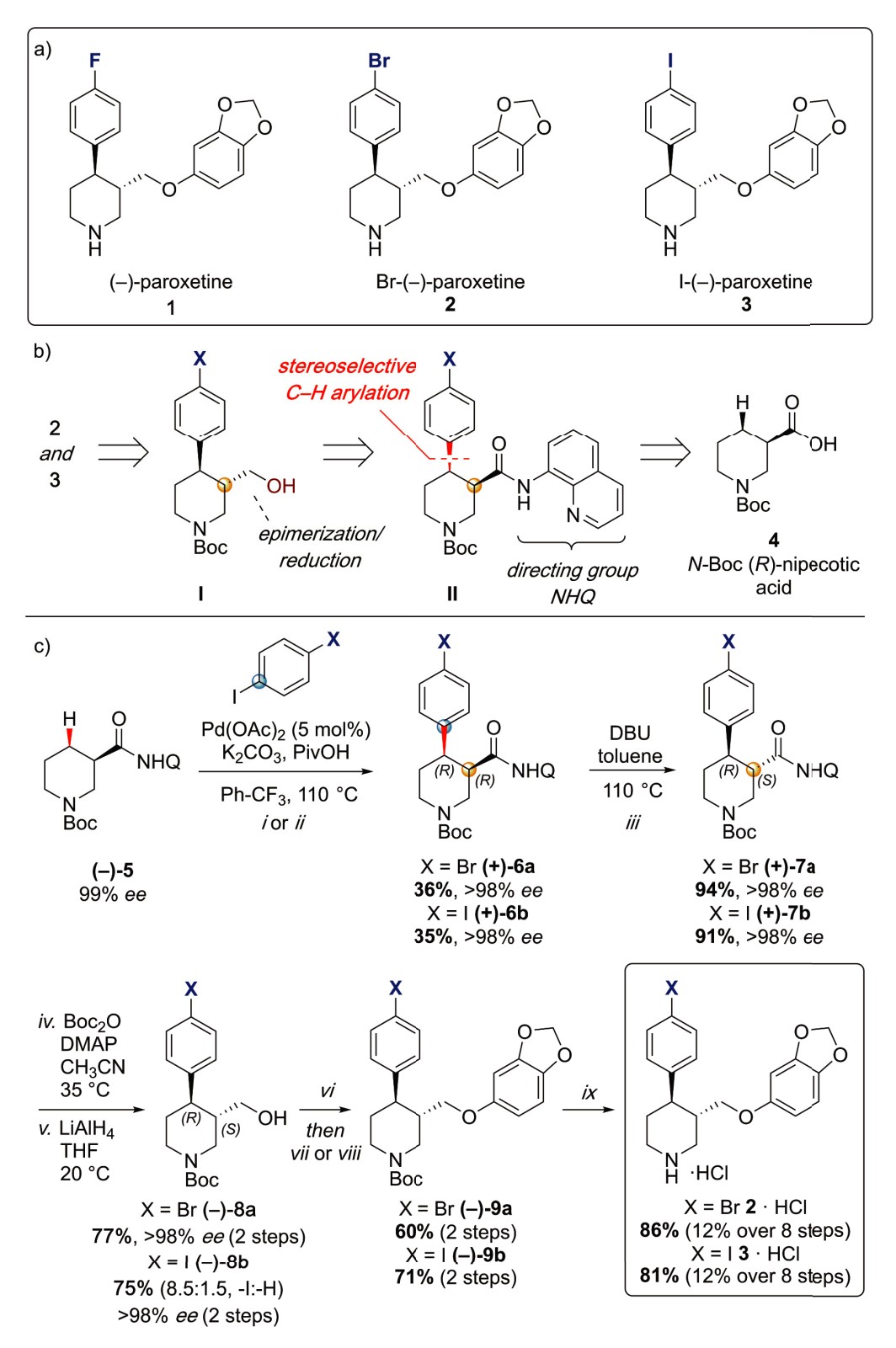

**Figure 2.** Synthesis of paroxetine analogues. (**a**) Structures of (–)-paroxetine (1) and the targeted Br- (2) and I-analogues (3). (**b**) Retrosynthetic analysis of Br- and I-(–)-paroxetine. (**c**) Synthesis of Br- and I-(–)-paroxetine 2 and 3. Q = 8 quinolinyl-. Reaction conditions: *i*) X = Br: (–)−5 (4.0 mmol), 4-bromo iodobenzene (three equiv), Pd(OAc)$_2$ (5 mol %), K$_2$CO$_3$ (one equiv), PivOH (one equiv), Ph-CF$_3$ (2 mL, 2 M), 110°C, 18 hr; *ii*) X = I: (–)−5 (4.0 mmol), 1,4-diiodobenzene (four equiv), Pd(OAc)$_2$ (5 mol %), K$_2$CO$_3$ (one equiv), PivOH (one equiv), Ph-CF$_3$ (2 mL, 2 M), 110°C, 18 hr; *iii*) DBU (three equiv), toluene

*Figure 2 continued on next page*

**Figure 2 continued**

(1 M), 110℃, 24 hr; *iv*) Boc$_2$O (four equiv), DMAP (20 mol %), CH$_3$CN (0.5 M), 35℃, 22 hr; *v*) LiAlH$_4$ (two equiv), THF, 20℃, 0.5 hr; *vi*) MsCl (1.3 equiv), Et$_3$N (1.4 equiv), CH$_2$Cl$_2$, 0 to 25℃, 2 hr; *vii*) X = Br: sesamol (1.6 equiv), NaH (1.7 equiv), THF, 0℃ to 70℃, 18 hr; *viii*) X = I: sesamol (2.0 equiv), NaH (2.2 equiv), DMF, 0℃ to 90℃, 20 hr; *ix*) 4 N HCl in dioxane (10 equiv), 0℃ to 25℃, 18 hr.

products, formed via a *trans*-palladacycle (*Antermite et al., 2018*), were also isolated (Appendix 1). Subsequent treatment with 1,8-diazabicyclo(5.4.0)undec-7-ene (DBU) gave complete C(3)-epimerization affording (+)−7a and (+)−7b with the desired *trans*-stereochemistry in 94% and 91% yields. The aminoquinoline group was removed through telescoped amide activation and reduction with LiAlH$_4$ at 20℃ to give enantiopure hydroxymethyl intermediates (−)−8a and (−)−8b in 77% and 75% yield. Mesylation and nucleophilic substitution with sesamol formed ether derivatives (−)−9a and (−)−9b, which were deprotected to give Br- and I-paroxetine 2 and 3. An overall yield of 12% over 8 steps from commercial material was obtained in both cases. At each stage, the identity of the products and purity was established by acquiring $^1$H and $^{13}$C nuclear magnetic resonance spectra, IR spectra, and by high-resolution mass spectrometry *Supplementary files 1* and *2*. Enantiopurity was assessed by high-performance liquid chromatography (HPLC) with reference to racemic or scalemic samples (*Supplementary file 1*).

We also employed several SERT variants and the 8B6 Fab in the biochemical and structural studies described here. The wild-type SERT construct used in transport experiments contains the full-length SERT sequence fused to a C-terminal GFP tag (*Table 1*). The ts2-active variant contains two thermostabilizing mutations (Ile291Ala, Thr439Ser) which allows for purification of the apo transporter for binding studies and has kinetics of 5-HT transport (K$_m$: 4.5 ± 0.6 μM, V$_{max}$: 21 ± 5 pmol min$^{-1}$) that are in a similar range as wild-type SERT (K$_m$: 1.9 ± 0.3 μM, V$_{max}$: 23 ± 1 pmol min$^{-1}$) (*Coleman et al., 2016a*; *Green et al., 2015*). The ts2-inactive variant (Tyr110Ala, Ile291Ala) (*Coleman and Gouaux, 2018*), by contrast, is unable to transport 5-HT but can be crystallized due to the stabilizing Tyr110Ala mutation (*Green et al., 2015*) and binds SSRIs with high-affinity. The ΔN72/ΔC13 SERT variant used for cryo-EM is otherwise wild-type SERT which has been truncated at the N- and C-termini (*Table 1*) and yet retains transport and ligand-binding activities (*Coleman et al., 2019*). Finally, the recombinant 8B6 Fab (*Coleman et al., 2016a*; *Coleman et al., 2016b*) was used to produce SERT-Fab complexes which were studied by X-ray crystallography and cryo-EM.

We began by assessing the functional effects of paroxetine, Br-paroxetine, and I-paroxetine on SERT activity by measuring their inhibition of 5-HT transport and *S*-citalopram competition binding. We assayed the ability of the Br- and I-paroxetine derivatives to inhibit 5-HT transport in HEK293 cells expressing wild-type SERT, observing that upon substituting the 4-fluoro group with 4-bromo or 4-iodo groups, the potency of inhibition of 5-HT transport in wild-type SERT decreased significantly from 4 ± 1 for paroxetine to 40 ± 20 for Br-paroxetine and 180 ± 70 nM for I-paroxetine (*Figure 3a*, *Table 2*). Next, we measured the binding of paroxetine, Br-paroxetine, and I-paroxetine

**Table 1.** Expression constructs used in this study.

| Name | Expression construct | Experiment |
|---|---|---|
| Wild-type SERT | Full-length human SERT with a C-terminal thrombin-GFP-StrepII-His$_{10}$ tag. | [$^3$H] 5-HT transport assays |
| ΔN72/ ΔC13 SERT | Wild-type SERT modified by deletion of 72 residues on N-term and 13 residues on C-term | Cryo-electron microscopy |
| ts2-inactive | Full-length SERT with thrombin cleavage sites inserted after Gln76 and Thr618 and carrying the Tyr110Ala, Ile291Ala thermostabilizing mutations with additional mutations of surface-exposed cysteines Cys554, Cys580, and Cys622 to alanine | X-ray crystallography and [$^3$H] citalopram binding assays |
| ts2-active | Full-length SERT with thrombin cleavage sites inserted after Gln76 and Thr618 and carrying the Ile291Ala, Thr439Ser thermostabilizing mutations with additional mutations of surface-exposed cysteines Cys554, Cys580, and Cys622 to alanine | [$^3$H] citalopram binding assays |
| Asn177 mutants | Asn177 mutated to either Val, Thr, or Gln in ts2-active background | [$^3$H] citalopram binding assays |

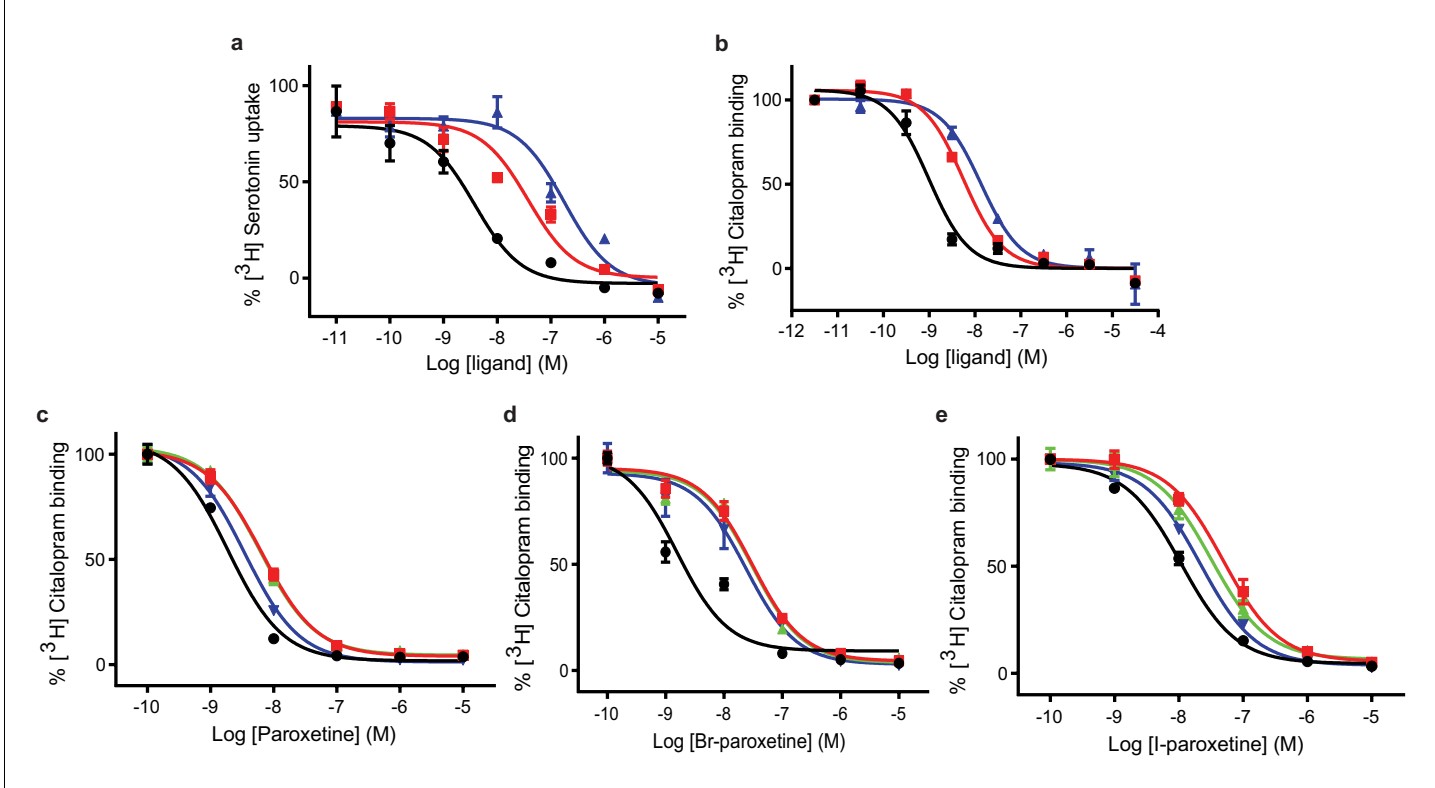

**Figure 3.** Inhibition of [³H]5-HT transport and [³H]citalopram binding by paroxetine and the Br- and I-derivatives. (a) 5-HT-transport of wild-type SERT and its inhibition by paroxetine, Br-, and I-paroxetine. Data are mean ± s.e.m. (n = 6). (b) Competition binding of paroxetine and its derivatives to ts2-inactive SERT. In panels a and b, paroxetine, Br-paroxetine, and I-paroxetine curves are shown as black, red, and blue lines, respectively. Data are mean ± s.e.m. (n = 6). (c) Competition binding of paroxetine to ts2-active (black), Asn177Val (red), Asn177Thr (green), and Asn177Gln (blue). Data are mean ± s.e.m. (n = 3). (d) Competition binding of Br-paroxetine. Data are mean ± s.e.m. (n = 3). (e) Competition binding of I-paroxetine. Data are mean ± s.e.m. (n = 3). The values associated with these experiments are reported in *Tables 2* and *3*.

to ts2-active and ts2-inactive SERT using *S*-citalopram competition binding assays, finding that the SERT variants employed in this study exhibited high-affinity for paroxetine and its derivatives (*Table 3*). A decrease in the binding affinity upon substituting the 4-fluoro group of paroxetine with 4-bromo or 4-iodo groups was observed in the competition binding assays. However, the difference in the binding affinities between paroxetine variants measured by the competition binding assay was not as pronounced as the difference in the inhibition potencies observed in the 5-HT transport assays (*Tables 2* and *3*). For example, the ts2-inactive (Tyr110Ala, Ile291Ala) variant employed in the previous (*Coleman and Gouaux, 2018*) and present x-ray studies exhibited a $K_i$ of 0.17 ± 0.02 nM for paroxetine, 0.94 ± 0.01 nM for Br-paroxetine, and a further decrease in affinity to I-paroxetine (2.3 ± 0.1 nM). The ts2-active SERT variant binds with similar affinity to paroxetine and Br-paroxetine, and shows a 4–5 fold decrease in affinity to I-paroxetine (*Figure 3b*, *Table 3*).

In the x-ray structures of SERT, paroxetine was modeled in the ABC pose such that the benzo-dioxol group is in subsite B (*Coleman and Gouaux, 2018*; *Coleman et al., 2016a*). A recent study suggested that binding affinity and potency to inhibit the transport of Br-paroxetine was only

**Table 2.** Inhibition of 5-HT transport by paroxetine and its derivatives.

| Ligand | IC₅₀ |
|---|---|
| Paroxetine | 4 ± 1 nM |
| Br-paroxetine | 40 ± 20 nM |
| I-paroxetine | 0.18 ± 0.07 µM |

**Table 3.** Binding of paroxetine and its derivatives to SERT variants used in this study.

| SERT variant | $K_i$ (nM) | | |
| --- | --- | --- | --- |
| | Paroxetine | Br-paroxetine | I-paroxetine |
| ts2-inactive | 0.17 ± 0.02 | 0.94 ± 0.01 | 2.3 ± 0.1 |
| ts2-active | 0.31 ± 0.07 | 0.4 ± 0.2 | 1.7 ± 0.3 |
| Asn177Val | 1.11 ± 0.04 | 5 ± 1 | 7.3 ± 0.9 |
| Asn177Thr | 1.0 ± 0.1 | 5 ± 2 | 4.4 ± 0.4 |
| Asn177Gln | 0.58 ± 0.07 | 4 ± 1 | 3.6 ± 0.4 |

moderately affected upon mutating a non-conserved residue Ala169 to Asp in subsite B of SERT (*Slack et al., 2019*; *Figure 1b*). We recently also identified a conserved residue, Asn177 in the sub-site B, which upon mutation exhibited differential effects on the inhibitory potency of ibogaine and noribogaine (*Coleman et al., 2019*). To further probe the role of Asn177 in subsite B, we studied the binding of paroxetine and its derivatives to selected Asn177 mutants designed in the ts2-active background (*Figure 1b*). We observed that the affinity of paroxetine to ts2-active SERT decreased by three-fold when Asn177 is substituted with small non-polar or polar residues such as valine and threonine, while only a 2-fold change in $K_i$ was observed for glutamine (Asn177Gln) (*Figure 3c*). In the case of Br-paroxetine, the Asn177 variants ($K_i$ between 4 and 5 nM) display up to a 10–13 fold decrease in $K_i$ when compared with ts2-active SERT (0.4 ± 0.2 nM) (*Figure 3d*, *Table 3*). The Asn177 variants show 2–4 fold decrease in affinity to I-paroxetine, with ts2-active SERT exhibiting a $K_i$ of 1.7 ± 0.3 nM and the mutants a $K_i$ of 4–7 nM. In the case of all three paroxetine variants, the reduction in affinity was the lowest for glutamine substitution. Irrespective of the SERT variant used, substitution of fluoro group with bromo or iodo group invariably decreased the affinity of paroxetine (*Figure 3e*, *Table 3*).

To define the binding poses of paroxetine and its analogues to SERT, we solved the structures of the ΔN72/ΔC13 and the ts2-inactive SERT variants complexed with Br- and I-paroxetine using single particle cryo-EM and X-ray crystallography (*Figure 4—figure supplements 1* and *2*). We began by collecting cryo-EM data sets for ΔN72/ΔC13 SERT-8B6 Fab complexes with each ligand. The TM densities in all three reconstructions were well-defined and contiguous allowing for clear positioning of the main chain in an outward-open conformation (*Figure 4—figure supplements 3* and *4*). Large aromatic side-chains were well-resolved for all three complexes, also suggesting that the aromatic moieties of paroxetine and its analogues could be identified and positioned in our cryo-EM maps. In addition, the particle distribution and orientations of SERT-Fab complexes in presence of Br- and I-paroxetine were similar to paroxetine, allowing for uniform comparison between the maps.

The ~ 3.3 Å resolution map of the ΔN72/ΔC13 SERT-8B6 paroxetine complex allowed us to locate a density feature for the inhibitor at the central site (*Figure 4a*). The resolution of the Br- and I-paroxetine complexes was comparatively lower at ~ 4.1 Å and ~ 3.8 Å, respectively (*Table 4*, *Figure 4—figure supplement 4*). Nevertheless, these ligands could also be modeled into the density at the central site with a correlation coefficient (CC) of 0.75 and 0.77, respectively (*Figure 4b–e*). To compare paroxetine in the ABC *vs.* the ACB pose, we flexibly modeled paroxetine in both poses at the central site followed by real space refinement. We observed that in the ACB pose, paroxetine could be positioned with a CC of 0.70 compared with 0.84 for the ABC pose suggesting that while ABC pose is clearly preferred under the conditions we tested, the possibility of an ACB pose cannot be excluded (*Figure 4—figure supplement 5a,b*). Based on the higher CC value, and the binding pose information from the ts2-inactive and ts3 SERT x-ray structures, the density in cryo-EM maps for paroxetine at the central site was interpreted to best accommodate ABC pose (*Coleman and Gouaux, 2018*; *Coleman et al., 2016a*). We also compared the reconstructed complexes by calculating difference maps, attempting to identify features associated with the scattering of bromine and iodine at the central and allosteric sites. However, the resulting difference maps did not contain any interpretable difference densities and thus did not further assist in ligand modeling. In the cryo-EM maps, the maltose headgroup of a DDM molecule could also be visualized in the allosteric site with the detergent tail inserted between TMs 10, 11, and 12. In contrast, in the X-ray maps only the head group of the octyl-maltoside detergent could be modeled due to the weak density of the hydrocarbon chain.

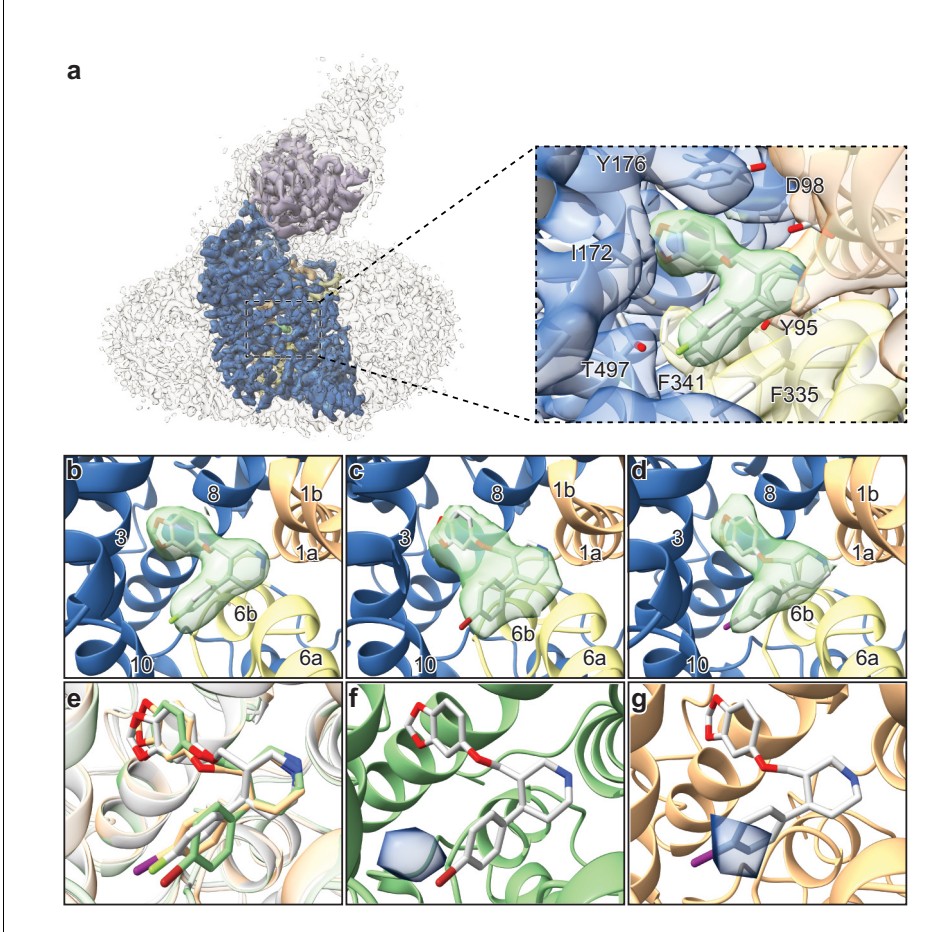

**Figure 4.** Structures of SERT-paroxetine complexes. (**a**) Cryo-EM reconstruction of SERT bound to paroxetine where the shape of the SERT-8B6 Fab complex and detergent micelle is shown in transparent light grey. The density of SERT is shown in dark blue with TM1 and TM6 colored in orange and yellow, respectively, and the density for paroxetine in green. The variable domain of the 8B6 Fab is colored in purple. Inset shows the density features at the central site of paroxetine. (**b**) Density feature at the central site of paroxetine. (**c**) Density feature at the central site of Br-paroxetine. (**d**) Density feature at the central site of I-paroxetine. (**e**) Comparison of the binding poses of paroxetine (grey), Br-paroxetine (green), and I-paroxetine (orange). (**f**) Anomalous difference electron density (blue) derived from Br-paroxetine, contoured at 5.2σ. g, Anomalous difference electron density (blue) derived from I-paroxetine, contoured at 4.3σ.

The online version of this article includes the following figure supplement(s) for figure 4:

**Figure supplement 1.** Work-flow of cryo-EM data processing of ΔN72/ΔC13 SERT/8B6 Fab/paroxetine complexes.

**Figure supplement 2.** 3D refinement of ΔN72/ΔC13 SERT/8B6 Fab/paroxetine complexes.

**Figure supplement 3.** Cryo-EM reconstruction of ΔN72/ΔC13 SERT/8B6 Fab/paroxetine complexes.

**Figure supplement 4.** Cryo-EM density segments of the transmembrane helices.

**Figure supplement 5.** Comparison of the fit of paroxetine in the ABC and ACB poses.

**Figure supplement 6.** Isomorphous difference densities at the central site.

We then explored the binding pose of paroxetine by growing crystals and collecting x-ray data of the ts2-inactive SERT-8B6 Fab complex with Br- and I-paroxetine (*Table 5*). Anomalous difference maps calculated from the previously determined ts2-inactive paroxetine structure (PDB ID: 6AWN) after refinement, showed clear densities for Br- and I- atoms of the paroxetine derivatives in subsite C (*Figure 4f,g*). No detectable anomalous peaks were observed in either subsite B or in the allosteric site and there were no other peaks in any other location above 2.5σ, suggesting that under these conditions, Br-paroxetine and I-paroxetine do not bind substantially in the ACB orientation or to the

**Table 4.** Cryo-EM data collection, refinement and validation statistics[a].

| | #1<br>(EMDB-21368)<br>(PDB 6VRH)<br>(EMPIAR-10380) | #2<br>(EMDB-21369)<br>(PDB 6VRK) | #3<br>(EMDB-21370)<br>(PDB 6VRL) |
|---|---|---|---|
| Data collection and processing | | | |
| Magnification | 77,160 | 77,160 | 77,160 |
| Voltage (kV) | 300 | 300 | 300 |
| Electron exposure (e–/Å$^2$) | 54–60 | 54 | 54 |
| Defocus range (μm) | −0.8 to −2.2 | −0.8 to −2.2 | −0.8 to −2.2 |
| Pixel size (Å) | 0.648 | 0.648 | 0.648 |
| Symmetry imposed | C1 | C1 | C1 |
| Initial particle images (no.) | 4,147,084 | 4,545,318 | 4,470,768 |
| Final particle images (no.) | 420,373 | 503,993 | 414,091 |
| Map resolution (Å)<br>FSC threshold | 3.3<br>0.143 | 4.1<br>0.143 | 3.8<br>0.143 |
| Map resolution range (Å)[†] | 4.25–3.25 | 5.75–3.75 | 5.50–3.50 |
| Refinement | | | |
| Initial model used (PDB code) | 6AWN | 6VRH | 6VRH |
| Initial model CC<br>Model resolution (Å)[‡]<br>FSC threshold | 0.64<br>3.7<br>0.5 | 0.70<br>4.3<br>0.5 | 0.71<br>4.1<br>0.5 |
| Model resolution range (Å) | 25.9–3.3 | 33.0–4.1 | 29.6–4.2 |
| Map sharpening $B$ factor (Å$^2$) | −85 | −174 | −161 |
| Model composition<br>Non-hydrogen atoms<br>Protein residues<br>Ligands (atoms) | 6143<br>764<br>254 | 6142<br>764<br>254 | 6142<br>764<br>254 |
| $B$ factors (Å$^2$)<br>Protein<br>Ligand | 138<br>129 | 138<br>113 | 122<br>112 |
| R.m.s. deviations<br>Bond lengths (Å)<br>Bond angles (°) | 0.002<br>0.48 | 0.002<br>0.59 | 0.002<br>0.54 |
| Validation<br>Refined model CC<br>MolProbity score<br>Clashscore<br>Poor rotamers (%) | 0.73<br>1.86<br>9.67<br>0 | 0.74<br>1.96<br>10.26<br>0 | 0.75<br>1.88<br>10.59<br>0.00 |
| Ramachandran plot<br>Favored (%)<br>Allowed (%)<br>Disallowed (%) | 94.84<br>5.16<br>0 | 93.54<br>6.46<br>0 | 95.12<br>4.88<br>0 |

[a]Data set #1 is the paroxetine reconstruction, #2 is Br-paroxetine, #3 I-paroxetine.

[†]Local resolution range.

[‡]Resolution at which FSC between map and model is 0.5.

allosteric site. Next, we calculated isomorphous difference maps ($F_o$-$F_o$) using the ts2-inactive paroxetine dataset (PDB: 6AWN) and either the Br-paroxetine or I-paroxetine datasets. The $F_o$(paroxetine)-$F_o$(Br-paroxetine) map also revealed a difference peak in subsite C near the halogenated groups while no significant peaks were detected in subsite B (*Figure 4—figure supplement 6a*). Similarly, the $F_o$(paroxetine)-$F_o$(I-paroxetine) map also contained a difference peak which overlapped with the position of the halogen (*Figure 4—figure supplement 6b*) while the $F_o$(Br-paroxetine)-$F_o$(I-paroxetine) difference map did not contain any interpretable features, likely due to the low resolution of both datasets (*Figure 4—figure supplement 6c*).

**Table 5.** X-ray data collection statistics.

| | Br-paroxetine (PDB 6W2B) | I-paroxetine (PDB 6W2C) |
|---|---|---|
| Data collection | | |
| Space group | C222$_1$ | C222$_1$ |
| Cell dimensions | | |
| a, b, c (Å) | 128.0, 161.9, 139.7 | 127.7, 161.9, 140.8 |
| α, β, γ (°) | 90, 90, 90 | 90, 90, 90 |
| Resolution (Å) | 20.45–4.69 (4.82–4.69)* | 25.98–6.12 (6.30–6.12)* |
| $R_{merge}$ | 13.60 (339.3) | 7.9 (292.9) |
| I / σI | 5.51 (0.49) | 5.01 (0.32) |
| CC$_{1/2}$ | 99.9 (16.5) | 99.8 (20.0) |
| Completeness (%) | 99.2 (100.0) | 92.6 (89.7) |
| Redundancy | 6.8 (6.2) | 1.8 (1.7) |

*Values in parentheses are for highest-resolution shell.

We next compared the cryo-EM structure of the SERT-paroxetine complex to the X-ray structure of the ts3 SERT paroxetine complex. Overall comparison of the transporter revealed only minor variation between structures solved by each method, with a Cα root-mean-square-deviation (RMSD) of 0.68 Å. The most significant differences between the cryo-EM and the X-ray structures were found at the extracellular and intracellular sites of TM12 and also in EL2, while the core of the transporter (TM1-10) was largely unchanged (*Figure 5a*). These changes can largely be explained on the basis of a crystal packing interface formed by TM12 and a highly flexible EL2 that is bound to the 8B6 Fab. We also compared central site residues involved in paroxetine binding, finding that the best fit to the cryo-EM density revealed only minor differences in the side-chains of Asp98, Tyr176, and Phe335 when compared to the x-ray structure (all atom RMSD: 0.91 Å) (*Figure 5b*). Finally, we compared the cryo-EM structures of the SERT 15B8 Fab/8B6 scFv paroxetine complex (PDB: 6DZW) to the SERT 8B6 Fab paroxetine complex to understand if these antibodies induce changes in transporter structure. Here we found that the most significant differences occurred in the extracellular domain and involved localized regions of EL2 and EL4 that interact with the antibody (*Figure 5c*). The transporter core was largely unchanged, with the only other significant differences being found in EL6, TM12, and IL4.

## Discussion

The binding of paroxetine to SERT has been extensively debated (*Coleman and Gouaux, 2018*; *Coleman et al., 2016a*; *Abramyan et al., 2019*; *Davis et al., 2016*; *Slack et al., 2019*). The first X-ray structure of the ts3-SERT variant demonstrated that the binding pose is such that the piperidine, benzodioxol, and fluorophenyl groups occupy subsites A, B, and C respectively, in the ABC pose (*Coleman et al., 2016a*; *Figure 1b*). Competition binding experiments using a variant of SERT containing a central binding site that has been genetically engineered to possess photo-cross-linking amino acids corroborated that paroxetine binds in a fashion which is similar to that observed in crystal structure (*Coleman and Gouaux, 2018*; *Coleman et al., 2016a*), where the fluorophenyl group is in proximity to Val501 (*Rannversson et al., 2017*). However, computational docking experiments using wild-type SERT predicted that the position of benzodioxol and fluorophenyl groups of paroxetine are 'flipped', with paroxetine occupying an ACB pose (*Davis et al., 2016*; *Figure 1c*). Subsequent studies involving wild-type and mutant SERT variants, that include modeling, mutagenesis, and Br-paroxetine docking experiments suggested that paroxetine could bind in both ABC and ACB poses. These studies also suggested that bromination of paroxetine and certain mutations near the central site, such as Ala169Asp, favored ABC pose (*Abramyan et al., 2019*; *Slack et al., 2019*). Hence, the authors in these studies hypothesized that the ABC pose observed in the crystal structure could be because of the crystallization conditions and thermostabilizing mutations.

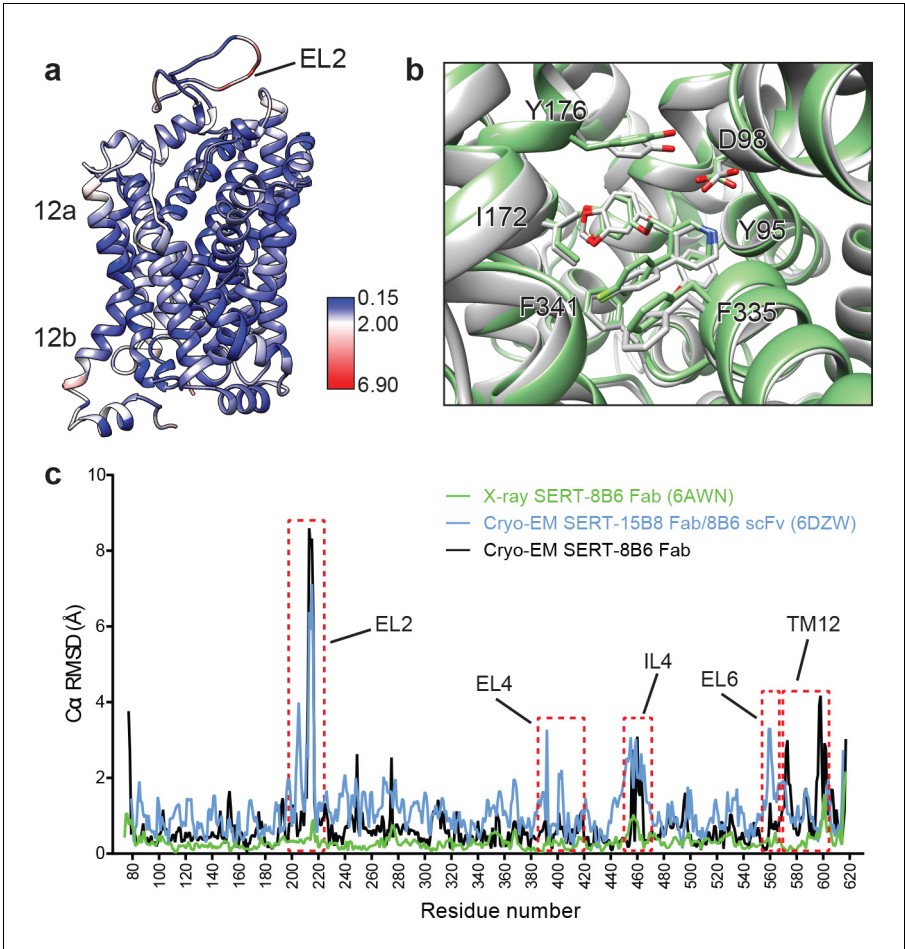

**Figure 5.** Comparison of the X-ray and cryo-EM structures of the SERT-paroxetine complex. (**a**) Superposition of the x-ray ts3-SERT-8B6 paroxetine structure (PDB: 5I6X) with the SERT-8B6 paroxetine complex determined by cryo-EM. The root-mean-square-deviations (RMSD) for Cα positions were plotted onto the cryo-EM SERT-8B6 paroxetine structure. (**b**) Comparison of the central binding site of the x-ray (grey) and cryo-EM (green) paroxetine structures. (**c**) The structure of the ts2-inactive SERT-8B6 scFv/15B8 Fab paroxetine (cryo-EM, 6DZW), ts2-inactive SERT-8B6 Fab paroxetine (x-ray, 6AWN), and the SERT-8B6 paroxetine (cryo-EM, this work) complexes were superposed onto the ts3 SERT-8B6 paroxetine complex (x-ray, 5I6X) as a reference. The RMSD for Cα positions were calculated for each structure in comparison with the reference. Regions with RMSD > 3.0 Å are shown boxed in red.

One of the thermostabilizing mutations in ts3-SERT, Thr439Ser, is near the central binding site and Thr439 participates in a hydrogen bonding network in subsite B that, in turn, includes the dioxol group of paroxetine. To probe the role of the Thr439Ser mutation in modulating the binding pose of paroxetine, we solved the X-ray structure of ts2-inactive (Tyr110Ala, Ile291Ala) SERT, wherein the residue at position 439 was the wild-type threonine. Paroxetine could be modeled in the ABC pose in the X-ray structure of ts2-inactive SERT (*Coleman and Gouaux, 2018*). MD simulations of ts2-inactive SERT suggested that the Thr439Ser mutation weakens the Na2 site. Furthermore, MD simulations and binding and uptake kinetics experiments using wild-type SERT in presence of paroxetine and a variant of paroxetine where in the 4-fluoro group is substituted with 4-bromo group suggested that the paroxetine binding pose in SERT could be ambiguous because of the pseudo symmetry of the paroxetine molecule. It was noted that paroxetine could occupy both ABC and ACB poses with almost equivalent preference. Upon substituting the 4-fluoro with a bulkier 4-bromo group, the ABC pose was favored (*Abramyan et al., 2019*; *Slack et al., 2019*).

Structural studies of SERT in complex with paroxetine and its analogues were thus required to resolve the uncertainty in paroxetine binding pose at the central site. Previously, we had demonstrated that cryo-EM can be used to define the position of ligands at the central site of SERT (*Coleman et al., 2019*). Here, we employed a similar methodology using the ΔN72/ΔC13 SERT variant complexed with 8B6 Fab to study binding of paroxetine at the central site. The density feature of paroxetine in the cryo-EM map at ~ 3.3 Å clearly resolved the larger benzodioxol and smaller fluorophenyl groups in subsite B and C, respectively (*Figure 4b*). Though this reconstruction suggests that paroxetine binds in the ABC pose, we also considered the possibility that the inhibitor density feature may represent an average of the ABC and ACB poses. We expected that if Br- and I-paroxetine were suitable surrogates for paroxetine, their binding pose would be unaffected by their reduced electronegativity and the size of the halogenated groups and therefore that they would also be associated with a comparable density feature at this site, as demonstrated by our cryo-EM maps. To further explore if there was a fraction of Br- or I-paroxetine in the ACB pose, we examined the position of the Br- or I- atoms at the central site by X-ray crystallography. If Br- and I-paroxetine were to bind in both the ABC or ACB poses, we expected to observe two anomalous peaks in our x-ray maps in subsites B and C; for both ligands, however, only a single detectable peak was observed in subsite C (*Figure 4f,g*). Thus, our direct biophysical observations reveal that under the conditions that we tested the ABC pose of paroxetine is preferred over the the ACB pose.

Paroxetine is stabilized at the central binding site by aromatic, ionic, non-ionic, hydrogen bonding, and cation-π interactions (*Coleman and Gouaux, 2018*). In the ABC pose, the amine of the piperidine ring of paroxetine binds with Asp98 (3.5 Å) and also makes a cation-π interaction with Tyr95 of subsite A (*Figure 4a*). The benzodioxol group of paroxetine, a catechol-like entity, occupies a position in subsite B which is similar to the binding of catechol derivative groups of sertraline and 3,4-dichlorophenethylamine in SERT (*Coleman and Gouaux, 2018*) and dDAT (*Wang et al., 2015a*) structures, respectively. In subsite B, the ring of Tyr176 makes an aromatic interaction with the benzodioxol while the hydrogen-bonding network in subsite B formed by Asn177, Thr439, backbone carbonyl oxygens, and amides are likely responsible for stabilization of the dioxol. The side-chain of Ile172 inserts between the benzodioxol and fluorophenyl, while the rings of Phe341 and Phe335 stack on either side of the fluorophenyl, 'sandwiching' it within subsite C. The halogen group of paroxetine and its analogues reside adjacent to the side-chain of Thr497 (4.0 Å), which may act to stabilize these groups through hydrogen bonding (*Figure 4a*). The larger atomic radius, the longer length of the carbon-halogen bond, and the difference in electronegativity of bromine (radius: 1.85 Å, bond-length: 1.92 Å, electronegativity: 2.96) and iodine (radius: 1.98 Å, bond-length: 2.14 Å, electronegativity: 2.66) relative to fluorine (radius: 1.47 Å, bond-length: 1.35 Å, electronegativity: 3.98) would explain why the fluorine analogue binds with greater affinity than Br-paroxetine and I-paroxetine.

We also explored the effect of conservative and non-conservative mutations in subsite B of SERT at Asn177 (*Figure 3*). Asn177 participates in a hydrogen-bond network with the hydroxyl group of noribogaine and with the dioxol of paroxetine. However, this network of interactions is also important for binding halogenated inhibitors in subsite B, as in the case for *S*-citalopram, fluvoxamine, and sertraline. All the mutants that we tested at Asn177 resulted in a loss of binding affinity to paroxetine and its analogues. Furthermore, the Ala169Asp mutation in subsite B (*Slack et al., 2019*; *Figure 1b,c*) also reduced paroxetine inhibition and binding, likely also disrupting these interactions. Although the effects were less severe when compared to paroxetine, Br-paroxetine binding and inhibition was also reduced for Ala169Asp (*Slack et al., 2019*). Thus, these mutations highlight the importance of subsite B interactions in paroxetine binding but they cannot be used to demonstrate the inhibitor pose because, in the ABC or ACB poses, either the dioxol or fluorine of paroxetine could act as a hydrogen-bond acceptor in subsite B.

Using a combination of chemical biology, cryo-EM, and X-ray crystallography we observed that under the conditions that we studied, the SSRI paroxetine preferably occupies the ABC pose at the central site, where it is involved in numerous interactions. However, the data presented in the manuscript does not completely exclude the possibility of an ACB pose at the central site. Our studies of the mechanism of paroxetine binding to SERT provide a robust framework for the design of experiments to identify new highly specific small-molecule SERT inhibitors.

# Materials and methods

**Key resources table**

| Reagent type (species) or resource | Designation | Source or reference | Identifiers | Additional information |
|---|---|---|---|---|
| Gene (*Homo sapiens*) | Human serotonin transporter | cDNA | NCBI Reference Sequence: NP_001036.1 | Dr. Randy D. Blakely (Florida Atlantic university brain institute) |
| Cell line (*Homo sapiens*) | HEK293S GnTI- | ATCC | Cat # ATCC CRL-3022 | Used for expression of SERT (PMID:27929454) |
| Cell line (*Spodoptera frugiperda*) | SF9 cells | ATCC | Cat # ATCC CRL-1711 | Used in production of baculovirus for transduction, and SERT antibodies (PMID:27929454) |
| Antibody | Mouse monoclonal. Isotype IgG2a, kappa | OHSU VGTI, Monoclonal Antibody Core | | 8B6 |
| Transfected construct (human) | pEG BacMam | Gouaux lab | | PMID:25299155 |
| Affinity chromatography resin | Strep-Tactin Superflow high capacity resin | Iba life sciences | Cat#2-1208-500 | |
| Chemical compound, drug | n-dodecyl-β-D-maltoside | Anatrace | Cat # D310 | Detergent |
| Chemical compound, drug | n-octyl β-D-maltoside | Anatrace | Cat # O310 | Detergent |
| Chemical compound, drug | fluorinated octyl-maltoside | Anatrace | Cat # O310F | Detergent |
| Chemical compound, drug | Cholesteryl Hemisuccinate | Anatrace | Cat # CH210 | Lipid |
| Chemical compound, drug | 1-palmitoyl-2-oleoyl-sn-glycero-3-phosphocholine | Anatrace | Cat # P516 | Lipid |
| Chemical compound, drug | 1-palmitoyl-2-oleoyl-sn-glycero-3-phosphoethanolamine | Anatrace | Cat # P416 | Lipid |
| Chemical compound, drug | 1-palmitoyl-2-oleoyl-sn-glycero-3-phosphoglycerol | Anatrace | Cat # P616 | Lipid |
| Chemical compound, drug | Paroxetine hydrochloride hemihydrate | Sigma | Cat # P9623 | Inhibitor |
| Chemical compound, drug | [$^3$H]5-HT | PerkinElmer | Cat # NET1167250UC | Radiolabeled substrate |
| Chemical compound, drug | [$^3$H]citalopram | PerkinElmer | Cat # NET1039250UC | Radiolabeled inhibitor |

*Continued on next page*

*Continued*

| Reagent type (species) or resource | Designation | Source or reference | Identifiers | Additional information |
|---|---|---|---|---|
| Software, algorithm | XDS | PMID:20124692 | RRID:SCR_015652 | http://xds.mpimf-heidelberg.mpg.de/ |
| Software, algorithm | Phaser | PMID:24189240 | RRID:SCR_014219 | https://www.phaser.cimr.cam.ac.uk/index.php/Phaser_Crystallographic_Software |
| Software, algorithm | Phenix | PMID:22505256 | RRID:SCR_014224 | https://www.phenix-online.org/ |
| Software, algorithm | SerialEM | PMID:16182563 | RRID:SCR_017293 | http://bio3d.colorado.edu/SerialEM |
| Software, algorithm | MotionCor2 | PMID:28250466 | RRID:SCR_016499 | http://msg.ucsf.edu/em/software/motioncor2.html |
| Software, algorithm | CTFFIND4 | PMID:26278980 | RRID:SCR_016732 | https://grigorieflab.umassmed.edu/ctffind4 |
| Software, algorithm | DoG-Picker | PMID:19374019 | | http://emg.nysbc.org/redmine/projects/software/wiki/DoGpicker |
| Software, algorithm | cryoSPARC | PMID:28165473 | RRID:SCR_016501 | https://cryosparc.com/ |
| Software, algorithm | RELION | PMID:23000701 | RRID:SCR_016274 | http://www2.mrc-lmb.cam.ac.uk/relion |
| Software, algorithm | cisTEM | PMID:29513216 | RRID:SCR_016502 | https://cistem.org/ |
| Software, algorithm | UCSF-Chimera | PMID:15264254 | RRID:SCR_004097 | https://www.cgl.ucsf.edu/chimera/ |
| Software, algorithm | Coot | PMID:15572765 | RRID:SCR_014222 | https://www2.mrc-lmb.cam.ac.uk/personal/pemsley/coot |
| Software, algorithm | MolProbity | PMID:20057044 | RRID:SCR_014226 | http://molprobity.biochem.duke.edu/ |
| Other | R 2/2 200 mesh Au holey carbon grids | Electron Microscopy Sciences | Cat # Q2100AR2 | Cryo-EM grids |
| Other | Copper HIS-Tag YSI | PerkinElmer | Cat # RPNQ0096 | SPA beads |

## SERT expression and purification

The human SERT constructs used in this study were the wild-type, the N- and C-terminally truncated wild-type (ΔN72/ΔC13), ts2-inactive (Tyr110Ala, Ile291Ala), and ts2-active (Ile291Ala, Thr439Ser) (*Coleman and Gouaux, 2018*; *Coleman et al., 2016a*; *Green et al., 2015*; *Coleman et al., 2019*; *Coleman et al., 2016b*) proteins (*Table 1*). The Asn177 mutants were generated in the ts2-active background. The expression and purification of SERT was carried out as previously described with minor modifications (*Coleman and Gouaux, 2018*; *Coleman et al., 2016a*; *Coleman et al., 2019*; *Coleman et al., 2016b*), as described below. All SERT constructs were cloned into BacMam vector system to be expressed as C-terminal GFP fusion using baculovirus-mediated transduction of HEK293S GnTI⁻ cells. Cells were solubilized in 20 mM Tris pH 8 with 150 mM NaCl, containing 20 mM n-dodecyl-β-D-maltoside (DDM) and 2.5 mM cholesteryl hemisuccinate (CHS), followed by purification using Strep-Tactin affinity chromatography in 20 mM Tris pH 8 with 100 mM NaCl (TBS), 1 mM DDM, and 0.2 mM CHS.

For cryo-EM of the ΔN72/ΔC13 SERT, 1 mM 5-HT was added during solubilization and affinity purification to stabilize SERT. GFP was cleaved from SERT by digestion with thrombin and the SERT-8B6 complex was made as described in the previous paragraph. The complex was separated from free Fab and GFP by SEC in TBS containing 1 mM DDM and 0.2 mM CHS, and the peak fractions were concentrated to 4 mg/ml followed by addition of either 200 µM paroxetine, Br-paroxetine or I-paroxetine.

For crystallization, no ligands were added during purification of ts2-inactive SERT, and 5% glycerol and 25 µM lipid (1-palmitoyl-2-oleoyl-sn-glycero-3-phosphocholine, 1-palmitoyl-2-oleoyl-sn-glycero-3-phosphoethanolamine, and 1-palmitoyl-2-oleoyl-sn-glycero-3-phosphoglycerol at a molar ratio of 1:1:1) were included in all the purification buffers. Following affinity purification, the fusion protein was digested by thrombin and EndoH and combined with recombinant 8B6 Fab at a molar ratio of 1:1.2. The SERT-8B6 complex was isolated by size-exclusion chromatography (SEC) on a Superdex 200 column in TBS containing 40 mM n-octyl β-D-maltoside, 0.5 mM CHS. The SERT-8B6 Fab complex was concentrated to 2 mg/ml and 1 µM 8B6 Fab and 50 µM Br-paroxetine or I-paroxetine was added prior to crystallization.

## Synthesis of Br- and I-paroxetine

All reactions were carried out under an inert atmosphere (argon) with flame-dried glassware using standard techniques, unless otherwise specified. Anhydrous solvents were obtained by filtration through drying columns (THF, MeCN, CH$_2$Cl$_2$ and DMF) or used as supplied (α,α,α-trifluorotoluene). Reactions in sealed tubes were run using Biotage microwave vials (2–5 ml or 10–20 ml recommended volumes). Aluminum caps equipped with molded butyl/PTFE septa were used for reactions in α,α,α-trifluorotoluene and toluene. Simple butyl septa were used for reactions in other solvents. Chromatographic purification was performed using 230–400 mesh silica with the indicated solvent system according to standard techniques. Analytical thin-layer chromatography (TLC) was performed on precoated, glass-backed silica gel plates. Visualization of the developed chromatogram was performed by UV absorbance (254 nm) and/or stained with a ninhydrin solution in ethanol. HPLC analyses were carried out on an Agilent 1260 Infinity Series system, employing Daicel Chiracel columns, under the indicated conditions. The high-resolution mass spectrometry (HRMS) analyses were performed using electrospray ion source (ESI). ESI was performed using a Waters LCT Premier equipped with an ESI source operated either in positive or negative ion mode. The software used was MassLynx 4.1; this software does not account for the electron and all the calibrations/references are calculated accordingly, that is [M+H]$^+$ is detected and the mass is calibrated to output [M+H]. Melting points are uncorrected. Infrared spectra (FTIR) were recorded in reciprocal centimeters (cm$^{-1}$).

Nuclear magnetic resonance spectra were recorded on 400 or 500 MHz spectrometers. The frequency used to record the NMR spectra is given in each assignment and spectrum ($^1$H NMR at 400 or 500 MHz; $^{13}$C NMR at 101 MHz or 126 MHz). Chemical shifts for $^1$H NMR spectra were recorded in parts per million from tetramethylsilane with the residual protonated solvent resonance as the internal standard (CHCl$_3$: δ 7.27 ppm, (CD$_2$H)$_2$SO: δ 2.50 ppm, CD$_2$HOD: δ 3.31 ppm). Data was reported as follows: chemical shift (multiplicity [s = singlet, d = doublet, t = triplet, m = multiplet and br = broad], coupling constant, integration and assignment). *J* values are reported in Hz. All multiplet signals were quoted over a chemical shift range. $^{13}$C NMR spectra were recorded with complete proton decoupling. Chemical shifts were reported in parts per million from tetramethylsilane with the solvent resonance as the internal standard ($^{13}$CDCl$_3$: δ 77.0 ppm, ($^{13}$CD$_3$)$_2$SO: δ 39.5 ppm, $^{13}$CD$_3$OD: δ 49.0 ppm). Assignments of $^1$H and $^{13}$C spectra, as well as *cis*- or *trans*-configuration, were based upon the analysis of δ and *J* values, analogy with previously reported compounds (*Antermite et al., 2018*), as well as DEPT, COSY and HSQC experiments, where appropriate. All Boc containing compounds appeared as a mixture of rotamers in the NMR spectra at room temperature. In some cases, NMR experiments for these compounds were carried out at 373 K to coalesce the signals, which is indicated in parentheses where appropriate. For NMR analysis performed at room temperature, 2D NMR experiments (COSY and HSQC) are also presented when useful for the assignments. Observed optical rotation (α') was measured at the indicated temperature (T °C) and values were converted to the corresponding specific rotations $[\alpha]_D^T$ in deg cm$^2$g$^{-1}$, concentration (*c*) in g per 100 mL. Full details of the synthetic route, using enantiopure and racemic substrates are

provided in Appendix 1, and NMR spectra of all reaction intermediates, 2 and 3, and HPLC analysis are cataloged in *Supplementary files 1* and *2*.

## Crystallization

Crystals of ts2-inactive SERT-8B6 Fab complex were grown by hanging-drop vapor diffusion at 4°C at a ratio of 2:1 (v/v) protein:reservoir. Br-paroxetine crystals were grown using reservoir solution containing 50 mM Tris pH 8.5, 20 mM $Na_2(SO4)$, 20 mM $LiCl_2$, 36% PEG 400, and 0.5% 6-aminohexanoic acid. I-paroxetine crystals were grown using a reservoir solution containing 100 mM HEPES pH 7.5, 40 mM $MgCl_2$, and 32% PEG 400.

## X-ray data collection

Crystals were harvested and flash cooled in liquid nitrogen. Data was collected at the Advanced Photon Source (Argonne National Laboratory, beamline 24-ID-C). Data for Br-paroxetine was collected at a wavelength of 0.91840 Å and at 1.37760 Å for I-paroxetine.

## Anomalous difference maps

X-ray data sets were processed with XDS (*Kabsch, 2010*); Friedel pairs were allowed to have different intensities. Molecular replacement was performed with coordinates from the previously determined ts2-inactive SERT-paroxetine structure (Protein Data Bank (PDB) code: 6AWN) (*Coleman and Gouaux, 2018*) using PHASER (*Bunkóczi et al., 2013*). B-factors were refined using PHENIX (*Afonine et al., 2012*) followed by generating anomalous difference maps using the phases derived from the higher resolution structures. To maximize the signal-to-noise ratio of the Br-paroxetine anomalous difference density, the high-resolution phases were blurred with a B-factor of 500 with a high-resolution cutoff of 5.5 Å. Using these optimized parameters for the Fourier analysis of the Br-paroxetine diffraction data, we obtained an anomalous map with the largest difference peak being present at 6.0σ and the noise level estimated at ∼ 2.5σ. To maximize the signal-noise-ratio of the I-paroxetine anomalous difference density, a high-resolution and low-resolution cutoff of 6.3 and 30 Å was applied during the generation of the anomalous maps. Using these optimized parameters for the Fourier analysis of the I-paroxetine diffraction data, we obtained an anomalous map with the largest difference peak being present at 4.5σ and the noise level estimated at ∼ 2.5σ.

## $F_o$-$F_o$ isomorphous difference maps

Isomorphous difference ($F_o$-$F_o$) maps were calculated in PHENIX by analyzing isomorphous pairs of crystals. Difference maps were calculated using the previously determined ts2-inactive SERT-paroxetine dataset and PDB (6AWN) for phasing. High- and low-resolution cutoffs of 6.0 and 30.0 Å were applied for the $F_o$(paroxetine)- $F_o$(Br-paroxetine) map and cutoffs of 6.3 and 30.0 Å were used for the $F_o$(paroxetine)- $F_o$(I-paroxetine) and $F_o$(Br-paroxetine)- $F_o$(I-paroxetine) maps.

## Cryo-EM grid preparation

To promote the inclusion of particles in thin ice, 100 μM fluorinated octyl-maltoside (final concentration) from a 10 mM stock was added to SERT-8B6 complexes immediately prior to vitrification. Quantifoil holey carbon gold grids, 2.0/2.0 μm, size/hole space, 200 mesh) were glow discharged for 60 s at 15 mA. SERT-8B6 Fab complex (2.5 μl) was applied to the grid followed by blotting for 2 s in the vitrobot and plunging into liquid ethane cooled by liquid $N_2$.

## Cryo-EM data collection and processing

Images were acquired using the automated program SerialEM (*Mastronarde, 2005*) on a FEI Titan Krios transmission electron microscope, operating at 300 keV and equipped with a Gatan Image Filter with the slit width set to 20 eV. A Gatan K3 direct electron detector was used to record movies in super-resolution counting mode with a binned pixel size of 0.648 Å per pixel. The defocus values ranged from −0.8 to −2.2 μm. Exposures of 1.0–1.5 s were dose fractioned into 40 frames, resulting in a total dose of 54–60 $e^-$ $Å^{-2}$. Movies were corrected for beam-induced motion using MotionCor2 (*Zheng et al., 2017*) with 5 × 5 patching. The contrast transfer function (CTF) parameters for each micrograph was determined using ctffind4 (*Rohou and Grigorieff, 2015*) and particles were picked either using DoG-Picker (*Voss et al., 2009*) or blob-based picking in cryoSPARC (*Punjani et al.,*

*2017*). DoG or cryoSPARC picked particles were independently subjected to 3D classification against a low-resolution volume of the SERT-8B6 complex. After sorting, the DoG and cryoSPARC picked particles were combined in RELION (*Scheres, 2012*) and the duplicate picks were removed (particle picks that are less than 100 Å of one another were considered duplicates). Combined particles were further sorted using reference-free 2D classification in cryoSPARC, followed by refinement in RELION and further 3D classification. Particles were then re-extracted (box size 400, 0.648 Å per pixel) and subjected to non-uniform refinement in cryoSPARC. Local refinement was then performed in *cis*TEM (*Grant et al., 2018*) with a mask that excludes the micelle and Fab constant domain to remove low-resolution features. The high-resolution refinement limit was incrementally increased while maintaining a correlation of 0.95 or better until no improvement in map quality was observed. The resolution of the reconstructions was accessed using the Fourier shell correlation (FSC) criterion and a threshold of 0.143 (*Rosenthal and Henderson, 2003*). Map sharpening was performed using local sharpening in PHENIX.

### Cryo-EM model building and refinement

A starting model was generated by fitting the X-ray structure of SERT-8B6 Fab paroxetine complex (PDB code: 6AWN) into the cryo-EM reconstruction in Chimera (*Pettersen et al., 2004*). Several rounds of manual adjustment and rebuilding were performed in Coot (*Emsley and Cowtan, 2004*), followed by real space refinement in PHENIX. For cross-validation, the FSC curve between the refined model and half maps was calculated and compared to prevent overfitting. Molprobity was used to evaluate the stereochemistry and geometry of the structures (*Chen et al., 2010*).

### Radioligand binding and uptake assays

Competition binding experiments were performed using scintillation proximity assays (SPA) (*Green et al., 2015*; *Coleman et al., 2016b*). The assays contained ~ 10 nM SERT, 0.5 mg/ml Cu-Ysi beads in TBS with 1 mM DDM, 0.2 mM CHS, and 10 nM [$^3$H]citalopram and 0.01 nM–1 mM of the cold competitors. Experiments were measured in triplicate. The error bars for each data point represent the s.e.m. Ki values were determined with the Cheng–Prusoff equation (*Cheng and Prusoff, 1973*) in GraphPad Prism. Uptake was measured as described previously in 96-well plates with [$^3$H]5-HT diluted 1:100 with unlabeled 5-HT. After 24 hr, cells were washed into uptake buffer (25 mM HEPES-Tris, pH 7.0, 130 mM NaCl, 5.4 mM KCl, 1.2 mM CaCl$_2$, 1.2 mM MgSO$_4$, 1 mM ascorbic acid and 5 mM glucose) containing 0.001–10,000 nM of the inhibitor. [$^3$H]5-HT was added to the cells and uptake was stopped by washing cells rapidly three times with uptake buffer. Cells were solubilized with 1% Triton-X100, followed by the addition of 200 μl of scintillation fluid to each well. The amount of labelled 5-HT was measured using a MicroBeta scintillation counter. Data were fit to a sigmoidal dose-response curve.

## Acknowledgements

We thank L Vaskalis for assistance with figures and H Owen for help with manuscript preparation. We acknowledge the staff of the Northeastern Collaborative Access Team at the Advanced Photon Source. A portion of this research was supported by NIH grant U24GM129547 and performed at the PNCC at OHSU and accessed through EMSL (grid.436923.9), a DOE Office of Science User Facility sponsored by the Office of Biological and Environmental Research. We are particularly grateful to Bernard and Jennifer LaCroute for their generous support. This work was funded by the NIH (5R37MH070039). EG is an investigator of the Howard Hughes Medical Institute.

We gratefully acknowledge The Royal Society [University Research Fellowship, UF140161 (to JAB), URF Appointed Grant RG150444].

## Additional information

### Funding

| Funder | Grant reference number | Author |
| --- | --- | --- |
| National Institutes of Health | 5R37MH070039 | Eric Gouaux |

| Howard Hughes Medical Institute | | Eric Gouaux |
| --- | --- | --- |
| Royal Society | UF140161 | James A Bull |
| Royal Society | RG150444 | James A Bull |

The funders had no role in study design, data collection and interpretation, or the decision to submit the work for publication.

### Author contributions

Jonathan A Coleman, Conceptualization, Data curation, Formal analysis, Validation, Investigation, Visualization, Methodology, Writing - original draft, Writing - review and editing; Vikas Navratna, Data curation, Formal analysis, Investigation, Visualization, Methodology, Writing - original draft, Writing - review and editing; Daniele Antermite, Conceptualization, Visualization, Methodology, Writing - original draft, Writing - review and editing; Dongxue Yang, Data curation, Methodology, Writing - review and editing; James A Bull, Conceptualization, Resources, Data curation, Supervision, Funding acquisition, Validation, Investigation, Visualization, Methodology, Writing - original draft, Writing - review and editing; Eric Gouaux, Conceptualization, Resources, Supervision, Funding acquisition, Validation, Investigation, Methodology, Project administration, Writing - review and editing

### Author ORCIDs

Jonathan A Coleman (iD) https://orcid.org/0000-0003-0001-6195
Vikas Navratna (iD) https://orcid.org/0000-0001-8599-1461
James A Bull (iD) http://orcid.org/0000-0003-3993-5818
Eric Gouaux (iD) https://orcid.org/0000-0002-8549-2360

### Decision letter and Author response

Decision letter https://doi.org/10.7554/eLife.56427.sa1
Author response https://doi.org/10.7554/eLife.56427.sa2

## Additional files

### Supplementary files

• Supplementary file 1. HPLC Traces for Racemic, Scalemic and Enantioenriched Br-Piperidine Derivatives (±)-S2a, (±)-S3a, (+)-6a, (+)-7a, (–)-S3a, (–)-8a and (+)-S4a.

• Supplementary file 2. NMR Spectra for Novel Compounds.

• Transparent reporting form

### Data availability

The coordinates and associated volumes for the cryo-EM reconstruction of SERT 8B6 Fab paroxetine, Br-paroxetine, and I-paroxetine datasets have been deposited in the PDB (https://www.rcsb.org/) and Electron Microscopy Data Bank (EMDB; https://www.ebi.ac.uk/pdbe/emdb/) under the accession codes 6VRH and 21368, 6VRK and 21369, and 6VRL and 21370, respectively. The half maps for each dataset have also been deposited in the EMDB (https://www.ebi.ac.uk/pdbe/emdb/). The x-ray coordinates for Br-paroxetine and I-paroxetine have been deposited in the PDB (https://www.rcsb.org/) under accession codes 6W2B and 6W2C, respectively.

The following datasets were generated:

| Author(s) | Year | Dataset title | Dataset URL | Database and Identifier |
| --- | --- | --- | --- | --- |
| Coleman JA, Navratna V, Yang D | 2020 | Cryo-EM structure of the wild-type human serotonin transporter complexed with paroxetine and 8B6 Fab | https://www.rcsb.org/structure/6VRH | RCSB Protein Data Bank, 6VRH |
| Coleman JA, Navratna V, Yang D | 2020 | Cryo-EM structure of the wild-type human serotonin transporter | https://www.ebi.ac.uk/pdbe/entry/emdb/EMD- | Electron Microscopy Data Bank, 21368 |

| | | | | | |
|---|---|---|---|---|---|
| | | | complexed with paroxetine and 8B6 Fab | | 21368 |
| Coleman JA, Navratna V, Yang D | | 2020 | Cryo-EM structure of the wild-type human serotonin transporter complexed with Br-paroxetine and 8B6 Fab | https://www.rcsb.org/structure/6VRK | RCSB Protein Data Bank, 6VRK |
| Coleman JA, Navratna V, Yang D | | 2020 | Cryo-EM structure of the wild-type human serotonin transporter complexed with Br-paroxetine and 8B6 Fab | https://www.ebi.ac.uk/pdbe/entry/emdb/EMD-21369 | Electron Microscopy Data Bank, 21369 |
| Coleman JA, Navratna V, Yang D | | 2020 | Cryo-EM structure of the wild-type human serotonin transporter complexed with I-paroxetine and 8B6 Fab | https://www.rcsb.org/structure/6VRL | RCSB Protein Data Bank, 6VRL |
| Coleman JA, Navratna V, Yang D | | 2020 | Cryo-EM structure of the wild-type human serotonin transporter complexed with I-paroxetine and 8B6 Fab | https://www.ebi.ac.uk/pdbe/entry/emdb/EMD-21370 | Electron Microscopy Data Bank, 21370 |
| Coleman JA, Navratna V, Yang D | | 2020 | Anomalous bromine signal reveals the position of Br-paroxetine complexed with the serotonin transporter at the central site | https://www.rcsb.org/structure/6W2B | RCSB Protein Data Bank, 6W2B |
| Coleman JA, Navratna V, Yang D | | 2020 | Anomalous iodine signal reveals the position of I-paroxetine complexed with the serotonin transporter at the central site | https://www.rcsb.org/structure/6W2C | RCSB Protein Data Bank, 6W2C |

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

## Appendix 1

### Synthesis of paroxetine analogues

#### Reagents

Commercial reagents were used as supplied or purified by standard techniques where necessary

Pd(OAc)$_2$, 8-Aminoquinoline, 1-(*tert*-butoxycarbonyl)piperidine-3-carboxylic acid and (*R*)−1-(*tert*-butoxycarbonyl)piperidine-3-carboxylic acid were purchased from Fluorochem Ltd and used as supplied.

PivOH and α,α,α-trifluorotoluene were purchased from Sigma-Aldrich Company Ltd and used as supplied.

K$_2$CO$_3$ was purchased from Sigma-Aldrich Company Ltd and flame-dried before use as part of reaction set-up.

Purity: Pd(OAc)$_2$,>98%; PivOH, 99%; K$_2$CO$_3$,≥98% (powder, –325 mesh), α,α,α-trifluorotoluene, anhydrous,≥99%.

Racemic and enantioenriched substrates *tert*-butyl (±)−3-(quinoline-8-ylcarbamoyl)piperidine-1-carboxylate ((±)-S1) and *tert*-butyl (–)-(*R*)−3-(quinolin-8-ylcarbamoyl)piperidine-1-carboxylate ((–)−5) were prepared by amide coupling of commercially available 8-aminoquinoline and the corresponding carboxylic acid (1-(*tert*-butoxycarbonyl)piperidine-3-carboxylic acid and (*R*)−1-(*tert*-butoxycarbonyl)piperidine-3-carboxylic acid, respectively) according to our previously reported procedures (*Antermite et al., 2018*).

### Structures of additional compounds in appendix 1

**Appendix 1—chemical structure 1.** Full Synthetic Route to Racemic and Enantioenriched Br-Piperidine Derivatives (±)-S2a, (±)-S3a, (+)−6a, (+)−7a, (–)-S3a, (–)−8a, (+)-S4a, (–)−9a and Br-(–)-paroxetine 2. In order to evaluate the enantiomeric excess of key intermediates (+)−6a and (+)−7a by chiral HPLC, the C–H arylation with 4-bromo iodobenzene was performed on both racemic ((±)-S1) and enantioenriched (–)−5 piperidine amide substrates (*Scheme 1*). The racemic synthesis was performed on a 0.5 mmol scale according to our previously reported protocol, (*Antermite et al., 2018*) and afforded *cis*-arylated derivatives (±)-S2a in 34% (*Scheme 1a*). A minor *trans*-functionalized product (±)-S3a, formed via a *trans*-palladacycle, (*Antermite et al., 2018*) was also isolated in 14%.

C–H Arylation of enantioenriched substrate (–)−5 proceed smoothly on a 4.0 mmol scale, and *cis*- and *trans*-piperidine products (+)−6a and (–)-S3a were isolated as single enantiomers in very similar yields (Scheme S1b). Subsequent treatment of enantiopure *cis*-derivative (+)−6a with DBU at 100°C afforded the *trans*-diastereomer as the right-handed enantiomer (+)−7a in 94% yield.

**Appendix 1—scheme 1.** Synthetic sequence, including the Pd-catalyzed C(4)–H arylation step, to access racemic and enantioenriched *cis*- and *trans*-piperidine amide derivatives (±)-S2a, (±)-S3a, (+)−6a, (+)−7a and (–)-S3a. (**a**) C–H Arylation conditions: (±)-S1 (0.5 mmol, one equiv) Ph-CF$_3$ (500 µL, 1 M). (**b**) C–H Arylation conditions: (–)−5 (4.0 mmol, one equiv), Ph-CF$_3$ (2.0 mL, 2 M). The enantiomeric excess of alcohol intermediates (+)-S4a and (–)−8a was evaluated after aminoquinoline removal on both enantiomeric *trans*-derivatives (–)-S3a and (+)−7a (*Scheme 2*). No undesired debromination was observed for the reductive aminoquinoline removal, and enantiopure alcohols (+)-S4a and (–)−8a were obtained in 70% and 77% yield, respectively. No erosion of enantiopurity should be expected after this step, given the literature precedents on the synthesis of (–)-paroxetine (*Amat et al., 2000*) and the absence of acidic protons in the substrate. Therefore, the synthesis was continued exclusively on alcohol derivative (–)−8a. O-Alkylation and Boc-deprotection with HCl finally afforded enantiopure Br-(–)-paroxetine analogue two as the corresponding hydrochloride salt in 12% yield over eight steps from commercial material.

**Appendix 1—scheme 2.** Reductive aminoquinoline removal and final steps in the synthesis of Br-(−)-paroxetine 2. (**a**) AQ removal on enantiomerically pure *trans*-piperidine (−)-S3a (0.2 mmol, one equiv). (**b**) AQ removal on enantiomerically pure *trans*-piperidine (+)−7a (1.1 mmol, one equiv) and final steps in the synthesis of Br-(−)-paroxetine 2. Full Synthetic Route to Racemic and Enantioenriched I-Piperidine Derivatives (±)-S2b, (±)-S3b, (+)−6b, (+)−7b, (−)-S3b, (−)−8b, (+)-S4b, (−)−9b and I-(−)-paroxetine 3 Similarly to the Br-analogue, C–H arylation with 1,4-diiodobenzene was performed on both racemic ((±)-S1) and enantioenriched ((−)−5) piperidine amide substrates (*Scheme 3*). The reaction proceeded well on both substrates affording racemic *cis*- and *trans*-arylated products (±)-S2b and (±)-S3b in 35% and 19% yield, and enantioenriched *cis*- and *trans*-derivatives (+)−6b and (−)-S3b in 35% and 20% yield respectively.

**Appendix 1—scheme 3.** Synthetic sequence, including the Pd-catalyzed C(4)–H arylation step, to access racemic and enantioenriched *cis*- and *trans*-piperidine amide derivatives (±)-S2b, (±)-S3b, (+)−6b, (+)−7b and (–)-S3b. (**a**) C–H Arylation conditions: (±)-S1 (0.5 mmol, one equiv) Ph-CF$_3$ (500 µL, 1 M). (**b**) C–H Arylation conditions: (–)−5 (4.0 mmol, one equiv), Ph-CF$_3$ (2.0 mL, 2 M). Reductive aminoquinoline cleavage was again performed to access enantiomeric *trans*-piperidine alcohols (+)-S4b and (–)−8b (*Scheme 4*). In both cases, a small degree of LiAlH$_4$-mediated dehalogenation was observed, and an inseparable mixture of the desired product and 10–15% of deiodinated material was isolated. However, the contaminant could be effectively removed after O-Alkylation, affording the pure aryl ether derivative (–)−9b in 71% yield. Final HCl-mediated Boc deprotection formed the desired I-(–)-paroxetine three as the corresponding HCl salt in 81% yield (12% yield over eight steps from commercial material).

**Appendix 1—scheme 4.** Reductive aminoquinoline removal and final steps in the synthesis of I-(–)-paroxetine 3. (**a**) AQ removal on enantiomerically pure *trans*-piperidine (–)-S3b (0.2 mmol, one equiv). (**b**) AQ removal on enantiomerically pure *trans*-piperidine (+)−7b (1.0 mmol, one equiv) and final steps in the synthesis of I-(–)-paroxetine 3.

## Experimental details and characterization data

Synthesis of Br-analogue of (–)-paroxetine (compounds (±)-S2a, (±)-S3a, (+)−6a, (+)−7a, (–)-S3a, (–)−8a, (+)-S4a, (–)-9a and 2 • HCl).

**Appendix 1—chemical structure 2.** *tert*-Butyl *cis*-(±)−4-(4-bromophenyl)−3-(quinolin-8-ylcarbamoyl) piperidine-1-carboxylate ((±)-S2a) and *tert*-butyl *trans*-(±)−4-(4-bromophenyl)−3-(quinolin-8-ylcarba-moyl)piperidine-1-carboxylate ((±)-S3a).

A reaction tube was charged with K$_2$CO$_3$ (69.1 mg, 0.50 mmol, one equiv), flame-dried, and allowed to cool under argon. *tert*-Butyl (±)−3-(quinoline-8-ylcarbamoyl)piperidine-1-carboxylate ((±)-S1) (178 mg, 0.50 mmol, one equiv), 4-bromoiodobenzene (424 mg, 1.50 mmol, three equiv), Pd (OAc)$_2$ (5.60 mg, 25.0 μmol, 5 mol %) and PivOH (51.2 mg, 0.50 mmol, one equiv) were added

sequentially. The reaction vessel was sealed with an aluminum cap (with molded butyl/PTFE septa) and purged with argon, then anhydrous PhCF₃ (500 µL, 1.0 M) was added by syringe. The reaction tube was then placed in a preheated oil bath and stirred at 110°C for 18 hr. The reaction mixture was allowed to cool to rt and EtOAc (10 mL) was added. The resulting mixture was filtered through a pad of Celite, eluting with further EtOAc (2 × 10 mL). The solvent was removed under reduced pressure, and the crude material was purified by flash column chromatography (0% to 5% $CH_3CN$/$CH_2Cl_2$). The product containing fractions were combined and the solvent was removed under reduced pressure. $Et_2O$ (5 mL) and pentane (5 mL) were added and the solvent was removed under reduced pressure to afford the minor product *tert*–butyl *trans*-(±)−4-(4-bromophenyl)−3-(quinolin-8-ylcarbamoyl) piperidine-1-carboxylate (±)-S3a as a pale yellow solid (34.5 mg, 14%) followed by the major product *tert*-butyl *cis*-(±)−4-(4-bromophenyl)−3-(quinolin-8-ylcarbamoyl)piperidine-1-carboxylate (±)-S2a as an off-white solid (87.2 mg, 34%).

## Major ((±)-S2a)

$R_f$ 0.31 (5% $CH_3CN$/$CH_2Cl_2$); mp = 81–86°C (from $Et_2O$/pentane);

$\nu_{max}$ (film)/$cm^{-1}$ 3343 (NH), 2859, 1684 (C = O), 1521, 1484, 1423, 1364, 1323, 1245, 1163, 1006, 827, 790, 757;

$^1H$ NMR (500 MHz, $(CD_3)_2SO$, 373 K) δ 9.75 (br s, 1 hr, NH), 8.83 (dd, $J$ = 4.2, 1.7 Hz, 1 hr, $HC_{Ar}$), 8.45 (dd, $J$ = 7.7, 1.4 Hz, 1 hr, $HC_{Ar}$), 8.31 (dd, $J$ = 8.3, 1.7 Hz, 1 hr, $HC_{Ar}$), 7.60–7.53 (m, 2 hr, $HC_{Ar}$), 7.48 (t, $J$ = 7.9 Hz, 1 hr, $HC_{Ar}$), 7.40–7.34 (m, 2 hr, $HC_{Ar}$), 7.33–7.26 (m, 2 hr, $HC_{Ar}$), 4.42 (ddd, $J$ = 14.8, 3.6, 1.7 Hz, 1 hr, NC*H*HCHCO), 4.25 (ddt, $J$ = 13.1, 4.6, 2.3 Hz, 1 hr, NC*H*HCH₂), 3.36–3.28 (m, 2 hr, NCH*H*CHCO, CHCO), 3.16 (dt, $J$ = 12.2, 4.0 Hz, 1 hr, CHAr), 3.02–2.92 (m, 1 hr, NCH*H*CH₂), 2.68 (qd, $J$ = 12.4, 4.7 Hz, 1 hr, NCH₂C*H*H), 1.72 (dq, $J$ = 12.9, 3.2 Hz, 1 hr, NCH₂CH*H*), 1.25 (s, 9 hr, C(CH₃)₃);

$^{13}C$ NMR (126 MHz, $(CD_3)_2SO$, 373 K) δ 169.8 (C = O amide), 153.4 (C = O carbamate), 147.9 ($C_{Ar}$), 142.0 ($C_{Ar}$ quat), 137.6 ($C_{Ar}$ quat), 135.7 ($C_{Ar}$), 133.9 ($C_{Ar}$ quat), 130.3 (2 × $C_{Ar}$), 129.1 (2 × $C_{Ar}$), 127.2 ($C_{Ar}$ quat), 126.1 ($C_{Ar}$), 121.2 ($C_{Ar}$), 120.8 ($C_{Ar}$), 118.7 (Br$C_{Ar}$ quat), 115.7 ($C_{Ar}$), 77.9 ($C$(CH₃)₃), 46.2 (N$C$H₂CHCO), 45.6 ($C$HCO), 42.9 (N$C$H₂CH₂), 41.7 ($C$HAr), 27.4 (C($C$H₃)₃), 25.0 (NCH₂$C$H₂);

HRMS (ESI⁺) *m/z* Calculated for $C_{26}H_{29}N_3O_3^{79}Br$ [M+H] 510.1392; Found 510.1386.

SMILES: O = C([C@H]1CN(C(OC(C)(C)C)=O)CC[C@H]1C2 = CC = C(Br)C = C2)NC3 = C(N = CC = C4)C4 = CC = C3

InChI = 1S/C26H28BrN3O3/c1-26(2,3)33-25(32)30-15-13-20(17-9-11-19(27)12-10-17)21(16-30)24(31)29-22-8-4-6-18-7-5-14-28-23(18)22/h4-12,14,20–21H,13,15–16 H2,1–3 H3,(H,29,31)/t20-,21-/m0/s1

## Minor ((±)-S3a)

$R_f$ 0.41 (5% $CH_3CN$/$CH_2Cl_2$); mp = 77–83°C (from $Et_2O$/pentane);

$\nu_{max}$ (film)/$cm^{-1}$ 3340 (NH), 2926, 1677 (C = O), 1521, 1484, 1424, 1323, 1230, 1156, 1126, 999, 824, 757;

$^1H$ NMR (500 MHz, $(CD_3)_2SO$, 373 K) δ 9.73 (br s, 1 hr, NH), 8.85 (dd, $J$ = 4.2, 1.7 Hz, 1 hr, $HC_{Ar}$), 8.39 (dd, $J$ = 7.7, 1.4 Hz, 1 hr, $HC_{Ar}$), 8.31 (dd, $J$ = 8.3, 1.7 Hz, 1 hr, $HC_{Ar}$), 7.61–7.54 (m, 2 hr, $HC_{Ar}$), 7.47 (t, $J$ = 8.0 Hz, 1 hr, $HC_{Ar}$), 7.39–7.32 (m, 2 hr, $HC_{Ar}$), 7.34–7.28 (m, 2 hr, $HC_{Ar}$), 4.36 (ddd, $J$ = 12.9, 3.7, 1.8 Hz, 1 hr, NC*H*HCHCO), 4.13 (ddt, $J$ = 13.3, 4.3, 2.2 Hz, 1 hr, NC*H*HCH₂), 3.18–3.00 (m, 3 hr, NCH*H*CHCO, CHCO, CHAr), 2.99–2.90 (m, 1 hr, NCH*H*CH₂), 1.81 (dq, $J$ = 12.9, 2.8 Hz, 1 hr, NCH₂C*H*H), 1.66 (qd, $J$ = 12.8, 4.6 Hz, 1 hr, NCH₂CH*H*), 1.49 (s, 9 hr, C(CH₃)₃);

$^{13}C$ NMR (126 MHz, $(CD_3)_2SO$, 373 K) δ 169.8 (C = O amide), 153.4 (C = O carbamate), 148.0 ($C_{Ar}$), 142.4 ($C_{Ar}$ quat), 137.7 ($C_{Ar}$ quat), 135.7 ($C_{Ar}$), 133.5 ($C_{Ar}$ quat), 130.6 (2 × $C_{Ar}$), 129.1 (2 × $C_{Ar}$), 127.2 ($C_{Ar}$ quat), 126.1 ($C_{Ar}$), 121.4 ($C_{Ar}$), 121.3 ($C_{Ar}$), 118.9 (Br$C_{Ar}$ quat), 116.3 ($C_{Ar}$), 78.6 ($C$(CH₃)₃), 49.2 ($C$HCO), 46.2 (N$C$H₂CHCO), 43.9 ($C$HAr), 43.3 (N$C$H₂CH₂), 32.0 (NCH₂$C$H₂), 27.7 (C($C$H₃)₃);

HRMS (ESI⁺) *m/z* Calculated for $C_{26}H_{29}N_3O_3^{79}Br$ [M+H] 510.1392; Found 510.1382.

SMILES: O = C([C@@H]1CN(C(OC(C)(C)C)=O)CC[C@H]1C2 = CC = C(Br)C = C2)NC3 = C(N = CC = C4)C4 = CC = C3

InChI = 1S/C26H28BrN3O3/c1-26(2,3)33-25(32)30-15-13-20(17-9-11-19(27)12-10-17)21(16-30)24(31)29-22-8-4-6-18-7-5-14-28-23(18)22/h4-12,14,20–21H,13,15–16 H2,1–3 H3,(H,29,31)/t20-,21+/m0/s1

**Appendix 1—chemical structure 3.** *tert*-Butyl (+)-(*3R,4R*)−4-(4-bromophenyl)−3-(quinolin-8-ylcarbamoyl)piperidine-1-carboxylate ((+)−6a) and *tert*-butyl (–)-(*3R,4S*)−4-(4-bromophenyl)−3-(quinolin-8-ylcarbamoyl)piperidine-1-carboxylate ((–)-S3a).

A large microwave vial (10–20 mL recommended volume) was charged with $K_2CO_3$ (553 mg, 4.0 mmol, one equiv), flame-dried, and allowed to cool under argon. *tert*-Butyl (*R*)−3-(quinolin-8-ylcarbamoyl)piperidine-1-carboxylate (–)−5 (1.42 g, 4.0 mmol, one equiv), 4-bromoiodobenzene (3.40 g, 12.0 mmol, three equiv), $Pd(OAc)_2$ (45.1 mg, 0.2 mmol, 5 mol %) and PivOH (409 mg, 4.0 mmol, one equiv) were added sequentially. The reaction vessel was sealed with an aluminum cap (with molded butyl/PTFE septa) and purged with argon, then anhydrous $PhCF_3$ (2.0 mL, 2.00 M) was added by syringe. The reaction tube was then placed in a preheated oil bath and stirred at 110°C for 18 hr. The reaction mixture was then allowed to cool to rt and EtOAc (20 mL) was added. The resulting mixture was filtered through a pad of Celite, eluting with further EtOAc (2 × 50 mL). The solvent was removed under reduced pressure. The reaction mixture was purified by two consecutive chromatographic separations: one (0% to 5% $CH_3CN/CH_2Cl_2$) to isolate the minor *trans*-product *tert*-butyl (–)-(*3R,4S*)−4-(4-bromophenyl)−3-(quinolin-8-ylcarbamoyl)piperidine-1-carboxylate (–)-S3a followed by a second (10% to 15% acetone/pentane) to isolate the major *cis*-product *tert*-butyl (+)-(*3R,4R*)−4-(4-bromophenyl)−3-(quinolin-8-ylcarbamoyl)piperidine-1-carboxylate (+)−6a. The product containing fractions were combined and the solvent was removed under reduced pressure. $Et_2O$ (20 mL) and pentane (20 mL) were added and the solvent was removed under reduced pressure to afford the minor *trans*-product (–)-S3a as a pale yellow solid (371 mg, 18%, 98.0% ee) and the major *cis*-product (+)−6a as a white solid (730 mg, 36%, 98.2% ee).

## Major ((+)−6a)

$[\alpha]_D^{23}$ + 15.4 (c 1.3, CHCl₃).

Characterization data identical to that reported for racemic *cis*-piperidine (±)-S2a (see S17).

HPLC Conditions: Chiralpak IA 3-column, 85:15 *n*-hexane:*i*-PrOH, flow rate: 1 mL·min⁻¹, 35°C, UV detection wavelength: 210.4 nm. Retention times: 11.9 min (*3S,4S* enantiomer), 17.3 min (*3R,4R* enantiomer).

SMILES: O = C([C@H]1CN(C(OC(C)(C)C)=O)CC[C@H]1C2 = CC = C(Br)C = C2)NC3 = C(N = CC = C4)C4 = CC = C3

InChI = 1S/C26H28BrN3O3/c1-26(2,3)33-25(32)30-15-13-20(17-9-11-19(27)12-10-17)21(16-30)24(31)29-22-8-4-6-18-7-5-14-28-23(18)22/h4-12,14,20–21H,13,15–16 H2,1–3 H3,(H,29,31)/t20-,21-/m0/s1

## Minor ((–)-S3a)

– $[\alpha]_D^{23}$ 35.4 (c 1.3, CHCl₃).

Characterization data identical to that reported for racemic *trans*-piperidine (±)-S3a (see S17).

HPLC Conditions: Chiralpak IA 3-column, 85:15 *n*-hexane:*i*-PrOH, flow rate: 1 mL·min⁻¹, 35°C, UV detection wavelength: 254.1 nm. Retention times: 9.1 min (*3R,4S* enantiomer), 12.2 min (*3S,4R* enantiomer).

SMILES: O = C([C@H]1CN(C(OC(C)(C)C)=O)CC[C@@H]1C2 = CC = C(Br)C = C2)NC3 = C
(N = CC = C4)C4 = CC = C3.

InChI = 1S/C26H28BrN3O3/c1-26(2,3)33-25(32)30-15-13-20(17-9-11-19(27)12-10-17)21(16-30)24
(31)29-22-8-4-6-18-7-5-14-28-23(18)22/h4-12,14,20–21H,13,15–16 H2,1–3 H3,(H,29,31)/t20-,21+/m1/
s1.

**Appendix 1—chemical structure 4.** *tert*-Butyl (+)-(*3S,4R*)−4-(4-bromophenyl)−3-(quinolin-8-ylcarba-moyl)piperidine-1-carboxylate ((+)−7a).

A flame-dried reaction tube was charged with *cis*-3,4-disubstituted piperidine (+)−6a (662 mg, 1.30 mmol, one equiv) and 1,8-diazabicyclo(5.4.0)undec-7-ene (DBU, 600 µL, 3.90 mmol, three equiv). The reaction vessel was sealed with an aluminum cap (with molded butyl/PTFE septa) and purged with argon, then anhydrous toluene (1.30 mL, 1.0 M) was added by syringe. The reaction tube was then placed in a preheated oil bath and stirred at 110°C for 24 hr. The reaction mixture was then allowed to cool to rt and $CH_2Cl_2$ (5 mL) and sat. aq. $NH_4Cl$ (5 mL) were added. The phases were separated, and the aqueous layer was extracted with $CH_2Cl_2$ (3 × 10 mL). The combined organic extracts were dried over $Na_2SO_4$ and filtered. The solvent was removed under reduced pressure. The reaction mixture was purified by flash column chromatography (15% acetone/pentane). The product containing fractions were combined and the solvent was removed under reduced pressure. $Et_2O$ (10 mL) and pentane (10 mL) were added and the solvent was removed under reduced pressure to afford amide *tert*-butyl (+)-(*3S,4R*)−4-(4-bromophenyl)−3-(quinolin-8-ylcarbamoyl) piperidine-1-carboxylate (+)−7a as a white solid (621 mg, 94%, 98.4% ee).

$[\alpha]_D^{23}$ + 52.0 (*c* 1.0, $CHCl_3$).

Characterization data identical to that reported for racemic *trans*-piperidine (±)-S3a (see S17).

HPLC Conditions: Chiralpak IA 3-column, 85:15 *n*-hexane:*i*-PrOH, flow rate: 1 mL·min⁻¹, 35°C, UV detection wavelength: 254.1 nm. Retention times: 9.1 min (*3R,4S* enantiomer), 12.2 min (*3S,4R* enantiomer).

SMILES: O = C([C@@H]1CN(C(OC(C)(C)C)=O)CC[C@H]1C2 = CC = C(Br)C = C2)NC3 = C
(N = CC = C4)C4 = CC = C3.

InChI = 1S/C26H28BrN3O3/c1-26(2,3)33-25(32)30-15-13-20(17-9-11-19(27)12-10-17)21(16-30)24
(31)29-22-8-4-6-18-7-5-14-28-23(18)22/h4-12,14,20–21H,13,15–16 H2,1–3 H3,(H,29,31)/t20-,21+/m0/
s1.

**Appendix 1—chemical structure 5.** *tert*-Butyl (+)-(*3R,4S*)−4-(4-bromophenyl)−3-(hydroxymethyl) piperidine-1-carboxylate ((+)-S4a).

A flame-dried reaction tube was charged with amide (–)-S3a (102 mg, 0.20 mmol, one equiv), followed by di-*tert*-butyl dicarbonate (Boc₂O, 175 mg, 0.80 mmol, four equiv) and 4-(dimethylamino) pyridine (DMAP, 4.9 mg, 0.04 mmol, 20 mol %). The reaction vessel was sealed with an aluminum

cap (with molded butyl septa) and purged with argon, then anhydrous MeCN (400 μL, 0.5 M) was added by syringe. The mixture was then stirred at 35°C for 22 hr. The reaction mixture was then allowed to cool to rt and sat. aq. NH$_4$Cl (1 mL) and CH$_2$Cl$_2$ (1 mL) were added. The phases were separated, and the aqueous layer was extracted with CH$_2$Cl$_2$ (3 × 5 mL). The combined organic extracts were dried over Na$_2$SO$_4$ and filtered. The solvent was removed under reduced pressure to afford the crude N-Boc protected piperidine derivative.

This crude was solubilized in anhydrous THF (800 μL, 0.2 M) and the resulting solution was added dropwise to a suspension of LiAlH$_4$ (15.2 mg, 0.40 mmol, two equiv) in anhydrous THF (200 μL, 2.0 M) at 0°C under argon atmosphere. The mixture was then stirred at 20°C for 30 min. The reaction mixture was then quenched by slow addition of sat. aq. NH$_4$Cl (2 mL) at 0°C and stirred at rt for 30 min. The resulting suspension was filtered through a pad of Celite, eluting with EtOAc (3 × 5 mL). The phases were separated, and the aqueous layer was extracted with EtOAc (3 × 5 mL). The combined organic extracts were dried over Na$_2$SO$_4$ and filtered. The solvent was removed under reduced pressure. Purification by flash column chromatography (10% to 20% acetone/hexane) afforded primary alcohol (+)-S4a as a yellow solid (52.0 mg, 70% over two steps, 98.1% ee).

$[\alpha]_D^{23}$ + 5.0 (c 0.8, CHCl$_3$).

R$_f$0.21 (20% acetone/hexane); mp = 49–54°C;

$\nu_{max}$ (film)/cm$^{-1}$3407 (OH), 2922, 1662 (C = O), 1476, 1424, 1230, 1159, 1129, 1059, 1006, 816, 769;

$^1$H NMR (400 MHz, CDCl$_3$, 298 K) δ 7.47–7.40 (m, 2 hr, HC$_{Ar}$), 7.11–7.05 (m, 2 hr, HC$_{Ar}$), 4.36 (br d, J = 13.2 Hz, 1 hr, NCHHCHCH$_2$OH), 4.20 (br s, 1 hr, NCHHCH$_2$), 3.43 (dd, J = 11.0, 3.1 Hz, 1 hr, CHHOH), 3.26 (dd, J = 11.0, 6.4 Hz, 1 hr, CHHOH), 2.87–2.62 (m, 2 hr, NCHHCHCH$_2$OH, NCHHCH$_2$), 2.59–2.47 (m, 1 hr, CHAr), 1.88–1.59 (m, 4 hr, CHCH$_2$OH, NCH$_2$CH$_2$, OH), 1.49 (s, 9 hr, C(CH$_3$)$_3$);

$^{13}$C NMR (101 MHz, CDCl$_3$, 298 K, observed as a mixture of rotamers) δ 154.8 (C = O), 142.8 (C$_{Ar}$ quat), 131.8 (2 × C$_{Ar}$), 129.1 (2 × C$_{Ar}$), 120.3 (BrC$_{Ar}$ quat), 79.7 (C(CH$_3$)$_3$), 62.9 (CH$_2$OH), 46.5 (br m, NCH$_2$CHCH$_2$OH), 44.2 and 43.6 (NCH$_2$CH$_2$, CHAr, CHCH$_2$OH), 33.8 (NCH$_2$CH$_2$), 28.5 (C(CH$_3$)$_3$);

HRMS (ESI$^+$) m/z Calculated for C$_{19}$H$_{27}$N$_2$O$_3$Na$^{79}$Br [M+CH$_3$CN+Na Adduct] 433.1103; Found 433.1110.

HPLC Conditions: Chiralpak ID 3-column, 90:10 n-hexane:i-PrOH, flow rate: 1 mL·min$^{-1}$, 35°C, UV detection wavelength: 210.4 nm. Retention times: 8.0 min (3R,4S enantiomer), 8.6 min (3S,4R enantiomer).

SMILES: BrC1 = CC = C([C@@H]2[C@@H](CO)CN(C(OC(C)(C)C)=O)CC2)C = C1.

InChI = 1S/C17H24BrNO3/c1-17(2,3)22-16(21)19-9-8-15(13(10-19)11–20)12-4-6-14(18)7-5-12/h4-7,13,15,20H,8–11 H2,1–3 H3/t13-,15-/m1/s1.

**Appendix 1—chemical structure 6.** tert-Butyl (–)-(3S,4R)−4-(4-bromophenyl)−3-(hydroxymethyl) piperidine-1-carboxylate ((–)−8a).

A flame-dried round-bottom flask was charged with amide (+)-7a (565 mg, 1.11 mmol, one equiv), followed by di-tert-butyl dicarbonate (Boc$_2$O, 969 mg, 4.44 mmol, four equiv) and 4-(dimethylamino)pyridine (DMAP, 26.9 mg, 0.22 mmol, 20 mol %). The reaction vessel was sealed with an aluminum cap (with molded butyl septa) and purged with argon, then anhydrous MeCN (3.7 mL) and anhydrous CH$_2$Cl$_2$ (0.5 mL) were added by syringe. The mixture (0.3 M) was then stirred at 35°C for 22 hr. The reaction mixture was then allowed to cool to rt and sat. aq. NH$_4$Cl (5 mL) and CH$_2$Cl$_2$ (5 mL) were added. The phases were separated, and the aqueous layer was extracted with CH$_2$Cl$_2$ (3 × 10 mL). The combined organic extracts were dried over Na$_2$SO$_4$ and filtered. The solvent was removed under reduced pressure to afford the crude N-Boc protected piperidine derivative.

This crude was solubilized in anhydrous THF (3.5 mL, 0.3 M) and the resulting solution was added dropwise to a suspension of LiAlH$_4$ (84.2 mg, 2.22 mmol, two equiv) in anhydrous THF (2.0 mL, 1.0 M) at 0°C under argon atmosphere. The mixture was then stirred at 20°C for 30 min. The reaction mixture was then quenched by slow addition of sat. aq. NH$_4$Cl (5 mL) at 0°C and stirred at rt for 30 min. The resulting suspension was filtered through a pad of Celite, eluting with EtOAc (3 × 10 mL). The phases were separated, and the aqueous layer was extracted with EtOAc (3 × 10 mL). The combined organic extracts were dried over Na$_2$SO$_4$ and filtered. The solvent was removed under reduced pressure. Purification by flash column chromatography (10% to 20% acetone/hexane) afforded primary alcohol (–)−8a as a white solid (316 mg, 77% over two steps, 98.1% *ee*).

– $[\alpha]_D^{23}$ 8.0 (*c* 1.0, CHCl$_3$).

Characterization data identical to that reported for enantiomeric alcohol (+)-S4a (see S20).

HPLC Conditions: Chiralpak ID 3-column, 90:10 *n*-hexane:*i*-PrOH, flow rate: 1 mL·min$^{-1}$, 35°C, UV detection wavelength: 210.4 nm. Retention times: 8.0 min (3*R*,4*S* enantiomer), 8.6 min (3*S*,4*R* enantiomer).

SMILES: BrC1 = CC = C([C@H]2[C@H](CO)CN(C(OC(C)(C)C)=O)CC2)C = C1.

InChI = 1S/C17H24BrNO3/c1-17(2,3)22-16(21)19-9-8-15(13(10-19)11−20)12-4-6-14(18)7-5-12/h4-7,13,15,20H,8−11 H2,1−3 H3/t13-,15-/m0/s1.

**Appendix 1—chemical structure 7.** *tert*-Butyl (3*S*,4*R*)−3-((benzo[*d*][1,3]dioxol-5-yloxy)methyl)−4-(4-bromophenyl)piperidine-1-carboxylate ((–)−9a).

Alcohol (–)−8a (280 mg, 0.76 mmol, one equiv) and triethylamine (147 μL, 1.10 mmol, 1.4 equiv) were added to a flame-dried round-bottom flask, dissolved in anhydrous CH$_2$Cl$_2$ (4.0 mL, 0.2 M) and cooled down to 0°C. Methanesulfonyl chloride (75 μL, 0.97 mmol, 1.3 equiv) was then added by Gilson pipette. After stirring 5 min at 0°C, the reaction mixture was stirred at 25°C for 2 hr, then diluted with CH$_2$Cl$_2$ (5 mL) and sat. aq. NaHCO$_3$ (5 mL). The phases were separated, and the aqueous layer was extracted with CH$_2$Cl$_2$ (3 × 10 mL). The combined organic extracts were dried over Na$_2$SO$_4$ and filtered. The solvent was removed under reduced pressure to afford the crude mesylated alcohol derivative.

NaH (60% dispersion in mineral oil, 51.8 mg, 1.30 mmol, 1.7 equiv) was added to a solution of sesamol (168 mg, 1.20 mmol, 1.6 equiv) in anhydrous THF (4.0 mL, 0.3 M) at 0°C. The mixture was then stirred at 25°C for 1 hr. A solution of the crude mesylated alcohol in anhydrous THF (5.0 mL, 0.1 M) was then added dropwise to this suspension. The resulting mixture was stirred at 70°C for 18 hr. The reaction mixture was then quenched by addition of H$_2$O (5 mL) and diluted with EtOAc (5 mL). The phases were separated, and the aqueous layer was extracted with EtOAc (4 × 10 mL). The combined organic extracts were dried over Na$_2$SO$_4$ and filtered. The solvent was removed under reduced pressure. Purification by flash column chromatography (5% acetone/pentane) afforded piperidine (–)−9a as a white solid (225 mg, 60% over two steps).

– $[\alpha]_D^{23}$ 36.0 (*c* 1.0, CHCl$_3$).

R$_f$0.20 (5% acetone/pentane); mp = 53–58°C;

$v_{max}$ (film)/cm$^{-1}$ 2915, 1685 (C = O), 1483, 1424, 1230, 1163, 1129, 1036, 928, 816, 769;

$^1$H NMR (400 MHz, CDCl$_3$, 298 K) δ 7.45–7.38 (m, 2 hr, HC$_{Ar}$), 7.10–7.03 (m, 2 hr, HC$_{Ar}$), 6.64 (d, *J* = 8.5 Hz, 1 hr, HC$_{Ar}$), 6.36 (d, *J* = 2.5 Hz, 1 hr, HC$_{Ar}$), 6.14 (dd, *J* = 8.5, 2.5 Hz, 1 hr, HC$_{Ar}$), 5.89 (s, 2 hr, OCH$_2$O), 4.44 (br s, 1 hr, NC*H*HCHCH$_2$OAr), 4.25 (br s, 1 hr, NC*H*HCH$_2$), 3.61 (dd, *J* = 9.4, 2.8 Hz, 1 hr, C*H*HOAr), 3.45 (dd, *J* = 9.4, 6.4 Hz, 1 hr, CH*H*OAr), 2.92–2.73 (br m, 2 hr, NCH*H*CHCH$_2$OAr, NCH*H*CH$_2$), 2.67 (td, *J* = 11.7, 3.9 Hz, 1 hr, C*H*Ar), 2.08–1.97 (br m, 1 hr,

C*H*CH₂OAr), 1.85–1.77 (br m, 1 hr, NCH₂C*H*H), 1.72 (td, *J* = 12.6, 4.3 Hz, 1 hr, NCH₂CH*H*), 1.50 (s, 9 hr, C(CH₃)₃);

$^{13}$C NMR (101 MHz, CDCl₃, 298 K) δ 154.7 (C = O), 154.2 (OC$_{Ar}$ quat), 148.1 (OC$_{Ar}$ quat), 142.4 (C$_{Ar}$ quat), 141.7 (OC$_{Ar}$ quat), 131.8 (2 × C$_{Ar}$), 129.1 (2 × C$_{Ar}$), 120.4 (BrC$_{Ar}$ quat), 107.8 (C$_{Ar}$), 105.5 (C$_{Ar}$), 101.1 (OCH₂O), 98.0 (C$_{Ar}$), 79.7 (*C*(CH₃)₃), 68.7 (*C*H₂OAr), 47.3 (br m, N*C*H₂CHCH₂OAr), 44.2 (N*C*H₂CH₂, *C*HAr), 41.7 (*C*HCH₂OAr), 33.7 (NCH₂*C*H₂), 28.4 (C(*C*H₃)₃);

HRMS (ESI$^+$) *m/z* Calculated for C₂₄H₂₉NO₅$^{79}$Br [M+H] 490.1229; Found 490.1240.

SMILES: BrC1 = CC = C([C@H]2[C@H](COC3 = CC(OCO4)=C4C = C3)CN(C(OC(C)(C)C)=O)CC2)C = C1.

InChI = 1S/C24H28BrNO5/c1-24(2,3)31-23(27)26-11-10-20(16-4-6-18(25)7-5-16)17(13-26)14-28-19-8-9-21-22(12-19)30-15-29-21/h4-9,12,17,20H,10–11,13-15H2,1–3 H3/t17-,20-/m0/s1.

**Appendix 1—chemical structure 8.** (3*S*,4*R*)−3-((Benzo[d][1,3]dioxol-5-yloxy)methyl)−4-(4-bromo-phenyl)piperidine-1-ium chloride (2 • HCl).

4 N HCl in 1,4-dioxane (500 μL, 2.00 mmol, 10 equiv) was added to a solution of *N*-Boc protected piperidine (–)−9a (98.1 mg, 0.20 mmol, one equiv) in 1,4-dioxane (500 μL, 0.4 M) at 0˚C under air. The solution was stirred at 25˚C for 18 hr, then an ice-cold 1:1 mixture of Et₂O/pentane (1 mL) was added and formation of a solid precipitate was observed. This was filtered and washed with further ice-cold Et₂O/pentane mixture (2 × 5 mL). The solid precipitate was dried under reduced pressure to afford (3*S*,4*R*)−3-((benzo[d][1,3]dioxol-5-yloxy)methyl)−4-(4-bromophenyl) piperidine-1-ium chloride 2 • HCl as an off-white solid (73.5 mg, 86%).

− [α]$_D^{23}$ 82.0 (*c* 1.0, MeOH); mp = 206–209 ˚C;

ν$_{max}$ (film)/cm$^{-1}$3317 (NH), 2926, 2687, 1484, 1182, 1103, 1033, 932, 846, 813, 787;

$^1$H NMR (400 MHz, CD₃OD, 298 K) δ 7.50–7.44 (m, 2 hr, HC$_{Ar}$), 7.24–7.18 (m, 2 hr, HC$_{Ar}$), 6.63 (d, *J* = 8.4 Hz, 1 hr, HC$_{Ar}$), 6.39 (d, *J* = 2.5 Hz, 1 hr, HC$_{Ar}$), 6.18 (dd, *J* = 8.5, 2.5 Hz, 1 hr, HC$_{Ar}$), 5.87–5.84 (m, 2 hr, OCH₂O), 3.71–3.62 (m, 2 hr, C*H*HOAr, NC*H*HCHCH₂OAr), 3.59–3.49 (m, 2 hr, CH*H*OAr, NC*H*HCH₂), 2.21–2.11 (m, 2 hr, NCH*H*CHCH₂OAr, NCH*H*CH₂), 3.03–2.91 (m, 1 hr, CHAr), 2.49–2.37 (m, 1 hr, C*H*CH₂OAr), 2.10–2.01 (m, 2 hr, NCH₂C*H₂*);

$^{13}$C NMR (101 MHz, CD₃OD, 298 K) δ 155.3 (OC$_{Ar}$ quat), 149.7 (OC$_{Ar}$ quat), 143.5 (C$_{Ar}$ quat), 142.4 (OC$_{Ar}$ quat), 133.0 (2 × C$_{Ar}$), 130.5 (2 × C$_{Ar}$), 122.0 (BrC$_{Ar}$ quat), 108.8 (C$_{Ar}$), 106.7 (C$_{Ar}$), 102.5 (OCH₂O), 98.9 (C$_{Ar}$), 69.0 (*C*H₂OAr), 47.7 (N*C*H₂CHCH₂OAr), 45.5 (N*C*H₂CH₂), 42.9 (*C*HAr), 40.6 (*C*HCH₂OAr), 31.4 (NCH₂*C*H₂);

HRMS (ESI$^+$) *m/z* Calculated for C₁₉H₂₁NO₃$^{79}$Br [M–Cl] 390.0705; Found 390.0698.

SMILES: BrC1 = CC = C([C@H]2[C@H](COC3 = CC(OCO4)=C4C = C3)CNCC2)C = C1 .Cl.

InChI = 1S/C19H20BrNO3.ClH/c20-15-3-1-13(2-4-15)17-7-8-21-10-14(17)11-22-16-5-6-18-19(9-16)24-12-23-18;/h1-6,9,14,17,21H,7–8,10-12H2;1H/t14-,17-;/m0./s1.

Synthesis of I-analogue of (–)-paroxetine (compounds (±)-S2b, (±)-S3b, (+)−6b, (+)−7b, (–)-S3b, (–)−8b, (+)-S4b, (–)-9b and 3 • HCl).

**Appendix 1—chemical structure 9.** *tert*-Butyl *cis*-(±)−4-(4-iodophenyl)−3-(quinolin-8-ylcarbamoyl) piperidine-1-carboxylate ((±)-S2b) and *tert*-butyl *trans*-(±)−4-(4-iodophenyl)−3-(quinolin-8-ylcarba-moyl)piperidine-1-carboxylate ((±)-S3b).

A reaction tube was charged with $K_2CO_3$ (69.1 mg, 0.50 mmol, one equiv), flame-dried, and allowed to cool under argon. *tert*-Butyl (±)−3-(quinoline-8-ylcarbamoyl)piperidine-1-carboxylate ((±)-S1) (178 mg, 0.50 mmol, one equiv), 1,4-diiodobenzene (660 mg, 2.00 mmol, four equiv), $Pd(OAc)_2$ (5.60 mg, 25.0 μmol, 5 mol %) and PivOH (51.2 mg, 0.50 mmol, one equiv) were added sequentially. The reaction vessel was sealed with an aluminum cap (with molded butyl/PTFE septa) and purged with argon, then anhydrous $PhCF_3$ (500 μL, 1.0 M) was added by syringe. The reaction tube was then placed in a preheated oil bath and stirred at 110°C for 18 hr. The reaction mixture was allowed to cool to rt and EtOAc (10 mL) was added. The resulting mixture was filtered through a pad of Cel-ite, eluting with further EtOAc (2 × 10 mL). The solvent was removed under reduced pressure, and the crude material was purified by flash column chromatography (0% to 5% $CH_3CN/CH_2Cl_2$). The product containing fractions were combined and the solvent was removed under reduced pressure. $Et_2O$ (5 mL) and pentane (5 mL) were added and the solvent was removed under reduced pressure to afford the minor product *tert*–butyl *trans*-(±)−4-(4-iodophenyl)−3-(quinolin-8-ylcarbamoyl) piperi-dine-1-carboxylate (±)-S3b as a pale yellow solid (52.2 mg, 19%) followed by the major product *tert*-butyl *cis*-(±)−4-(4-iodophenyl)−3-(quinolin-8-ylcarbamoyl)piperidine-1-carboxylate (±)-S2b as a pale yellow solid (97.9 mg, 35%).

## Major ((±)-S2b)

$R_f$ 0.30 (5% $CH_3CN/CH_2Cl_2$); mp = 91–95°C (from $Et_2O$/pentane);

$v_{max}$ (film)/cm$^{-1}$ 3343 (NH), 2926, 1685 (C = O), 1521, 1483, 1424, 1364, 1323, 1245, 1159, 1118, 1003, 824, 790, 757;

$^1$H NMR (500 MHz, $(CD_3)_2SO$, 373 K) δ 9.75 (br s, 1 hr, NH), 8.83 (dd, J = 4.2, 1.7 Hz, 1 hr, HC$_{Ar}$), 8.45 (dd, J = 7.6, 1.4 Hz, 1 hr, HC$_{Ar}$), 8.31 (dd, J = 8.3, 1.7 Hz, 1 hr, HC$_{Ar}$), 7.60–7.53 (m, 4 hr, HC$_{Ar}$), 7.48 (t, J = 8.0 Hz, 1 hr, HC$_{Ar}$), 7.19–7.12 (m, 2 hr, HC$_{Ar}$), 4.42 (ddd, J = 14.9, 3.7, 1.8 Hz, 1 hr, NC*H*HCHCO), 4.25 (ddt, J = 13.2, 4.7, 2.4 Hz, 1 hr, NC*H*HCH$_2$), 3.35–3.28 (m, 2 hr, NCH*H*CHCO, CHCO), 3.14 (dt, J = 12.4, 4.2 Hz, 1 hr, CHAr), 3.01–2.92 (m, 1 hr, NCH*H*CH$_2$), 2.67 (qd, J = 12.4, 4.6 Hz, 1 hr, NCH$_2$C*H*H), 1.71 (dq, J = 13.0, 3.4 Hz, 1 hr, NCH$_2$CH*H*), 1.25 (s, 9 hr, C(CH$_3$)$_3$);

$^{13}$C NMR (126 MHz, $(CD_3)_2SO$, 373 K) δ 169.8 (C = O amide), 153.4 (C = O carbamate), 147.9 (C$_{Ar}$), 142.5 (C$_{Ar}$ quat), 137.6 (C$_{Ar}$ quat), 136.3 (2 × C$_{Ar}$), 135.7 (C$_{Ar}$), 133.9 (C$_{Ar}$ quat), 129.3 (2 × C$_{Ar}$), 127.2 (C$_{Ar}$ quat), 126.1 (C$_{Ar}$), 121.2 (C$_{Ar}$), 120.8 (C$_{Ar}$), 115.7 (C$_{Ar}$), 90.9 (IC$_{Ar}$ quat), 77.9 (*C*(CH$_3$)$_3$), 46.2 (NCH$_2$CHCO), 45.6 (*C*HCO), 42.9 (NCH$_2$CH$_2$), 41.8 (*C*HAr), 27.4 (C(CH$_3$)$_3$), 25.0 (NCH$_2$CH$_2$);

HRMS (ESI$^+$) *m/z* Calculated for $C_{26}H_{29}N_3O_3{}^{127}I$ [M+H] 558.1254; Found 558.1260.

SMILES: O = C([C@H]1CN(C(OC(C)(C)C)=O)CC[C@H]1C2 = CC = C(I)C = C2)NC3 = C(N = CC = C4)C4 = CC = C3.

InChI = 1S/C26H28IN3O3/c1-26(2,3)33-25(32)30-15-13-20(17-9-11-19(27)12-10-17)21(16-30)24(31)29-22-8-4-6-18-7-5-14-28-23(18)22/h4-12,14,20−21H,13,15−16 H2,1−3 H3,(H,29,31)/t20-,21-/m0/s1.

## Minor ((±)-S3b)

$R_f$ 0.41 (5% $CH_3CN/CH_2Cl_2$); mp = 93–96°C (from $Et_2O$/pentane); $v_{max}$ (film)/cm$^{-1}$ 3336 (NH), 2922, 1677 (C = O), 1521, 1483, 1424, 1323, 1230, 1156, 1062, 1003, 824, 757;

$^1$H NMR (500 MHz, (CD$_3$)$_2$SO, 373 K) δ 9.73 (br s, 1 hr, NH), 8.85 (dd, $J$ = 4.2, 1.7 Hz, 1 hr, HC$_{Ar}$), 8.39 (dd, $J$ = 7.7, 1.3 Hz, 1 hr, HC$_{Ar}$), 8.31 (dd, $J$ = 8.3, 1.7 Hz, 1 hr, HC$_{Ar}$), 7.62–7.55 (m, 2 hr, HC$_{Ar}$), 7.55–7.51 (m, 2 hr, HC$_{Ar}$), 7.47 (t, $J$ = 8.0 Hz, 1 hr, HC$_{Ar}$), 7.19–7.14 (m, 2 hr, HC$_{Ar}$), 4.35 (ddd, $J$ = 12.8, 3.8, 1.8 Hz, 1 hr, NC$H$HCHCO), 4.12 (ddt, $J$ = 13.3, 4.4, 2.1 Hz, 1 hr, NC$H$HCH$_2$), 3.17–2.99 (m, 3 hr, NCH$H$CHCO, CHCO, CHAr), 2.98–2.90 (m, 1 hr, NCH$H$CH$_2$), 1.80 (dq, $J$ = 13.3, 3.0 Hz, 1 hr, NCH$_2$C$H$H), 1.65 (qd, $J$ = 12.7, 4.6 Hz, 1 hr, NCH$_2$CH$H$), 1.48 (s, 9 hr, C(CH$_3$)$_3$);

$^{13}$C NMR (126 MHz, (CD$_3$)$_2$SO, 373 K) δ 169.8 (C = O amide), 153.4 (C = O carbamate), 148.1 (C$_{Ar}$), 142.8 (C$_{Ar}$ quat), 137.7 (C$_{Ar}$ quat), 136.6 (2 × C$_{Ar}$), 135.7 (C$_{Ar}$), 133.5 (C$_{Ar}$ quat), 129.3 (2 × C$_{Ar}$), 127.2 (C$_{Ar}$ quat), 126.1 (C$_{Ar}$), 121.4 (C$_{Ar}$), 121.3 (C$_{Ar}$), 116.3 (C$_{Ar}$), 91.1 (IC$_{Ar}$ quat), 78.6 (C(CH$_3$)$_3$), 49.1 (CHCO), 46.2 (NCH$_2$CHCO), 44.0 (CHAr), 43.3 (NCH$_2$CH$_2$), 32.0 (NCH$_2$CH$_2$), 27.7 (C(CH$_3$)$_3$);

HRMS (ESI$^+$) $m/z$ Calculated for C$_{26}$H$_{29}$N$_3$O$_3^{127}$I [M+H] 558.1254; Found 558.1247.

SMILES: O = C([C@@H]1CN(C(OC(C)(C)C)=O)CC[C@H]1C2 = CC = C(I)C = C2)NC3 = C(N = CC = C4)C4 = CC = C3.

InChI = 1S/C26H28IN3O3/c1-26(2,3)33-25(32)30-15-13-20(17-9-11-19(27)12-10-17)21(16-30)24(31)29-22-8-4-6-18-7-5-14-28-23(18)22/h4-12,14,20–21H,13,15–16 H2,1–3 H3,(H,29,31)/t20-,21+/m0/s1.

**Appendix 1—chemical structure 10.** *tert*-Butyl (+)-(3R,4R)−4-(4-iodophenyl)−3-(quinolin-8-ylcarba-moyl)piperidine-1-carboxylate ((+)−6b) and *tert*-butyl (–)-(3R,4S)−4-(4-iodophenyl)−3-(quinolin-8-ylcarbamoyl)piperidine-1-carboxylate ((–)-S3b).

A large microwave vial (10–20 mL recommended volume) was charged with K$_2$CO$_3$ (553 mg, 4.0 mmol, one equiv), flame-dried, and allowed to cool under argon. *tert*-Butyl (R)−3-(quinolin-8-ylcarba-moyl)piperidine-1-carboxylate (–)−5 (1.42 g, 4.0 mmol one equiv), 1,4-diiodobenzene (5.28 g, 16.0 mmol, four equiv), Pd(OAc)$_2$ (45.1 mg, 0.2 mmol, 5 mol %) and PivOH (409 mg, 4.0 mmol, one equiv) were added sequentially. The reaction vessel was sealed with an aluminum cap (with molded butyl/ PTFE septa) and purged with argon, then anhydrous PhCF$_3$ (2.0 mL, 2.00 M) was added by syringe. The reaction tube was then placed in a preheated oil bath and stirred at 110°C for 18 hr. The reaction mixture was then allowed to cool to rt and EtOAc (20 mL) was added. The resulting mixture was filtered through a pad of Celite, eluting with further EtOAc (2 × 50 mL). The solvent was removed under reduced pressure. The reaction mixture was purified by two consecutive chromatographic separations: one (0% to 5% CH$_3$CN/CH$_2$Cl$_2$) to isolate the minor *trans*-product *tert*-butyl (–)-(3R,4S)−4-(4-iodophenyl)−3-(quinolin-8-ylcarbamoyl)piperidine-1-carboxylate (–)-S3b followed by a second (10% to 15% acetone/pentane) to isolate the major *cis*-product *tert*-butyl (+)-(3R,4R)−4-(4-iodophenyl)−3-(quinolin-8-ylcarbamoyl)piperidine-1-carboxylate (+)−6b. The product containing fractions were combined and the solvent was removed under reduced pressure. Et$_2$O (20 mL) and pentane (20 mL) were added and the solvent was removed under reduced pressure to afford the minor *trans*-product (–)-S3b as a pale orange solid (441 mg, 20%, 98.1% *ee*) and the major *cis*-prod-uct (+)−6b (775 mg, 35%, 98.2% *ee*).

## Major ((+)−6b)

$[\alpha]_D^{23}$ + 9.1 (c 1.1, CHCl$_3$).

Characterization data identical to that reported for racemic *cis*-piperidine (±)-S2b (see S24).

HPLC Conditions: Chiralpak IA 3-column, 85:15 *n*-hexane:*i*-PrOH, flow rate: 1 mL·min$^{-1}$, 35°C, UV detection wavelength: 210.4 nm. Retention times: 12.2 min (3S,4S enantiomer), 17.7 min (3R,4R enantiomer).

SMILES: O = C([C@H]1CN(C(OC(C)(C)C)=O)CC[C@H]1C2 = CC = C(I)C = C2)NC3 = C(N = CC = C4)C4 = CC = C3.

InChI = 1S/C26H28IN3O3/c1-26(2,3)33-25(32)30-15-13-20(17-9-11-19(27)12-10-17)21(16-30)24(31)29-22-8-4-6-18-7-5-14-28-23(18)22/h4-12,14,20–21H,13,15–16 H2,1–3 H3,(H,29,31)/t20-,21-/m0/s1.

## Minor ((–)-S3b)

– $[\alpha]_D^{23}$ 45.5 (c 1.1, CHCl$_3$).

Characterization data identical to that reported for racemic *trans*-piperidine (±)-S3b (see S24).

HPLC Conditions: Chiralpak IA 3-column, 85:15 *n*-hexane:*i*-PrOH, flow rate: 1 mL·min$^{-1}$, 35°C, UV detection wavelength: 254.1 nm. Retention times: 9.4 min (3*R*,4*S* enantiomer), 13.3 min (3*S*,4*R* enantiomer).

SMILES: O = C([C@H]1CN(C(OC(C)(C)C)=O)CC[C@@H]1C2 = CC = C(I)C = C2)NC3 = C(N = CC = C4)C4 = CC = C3.

InChI = 1S/C26H28IN3O3/c1-26(2,3)33-25(32)30-15-13-20(17-9-11-19(27)12-10-17)21(16-30)24(31)29-22-8-4-6-18-7-5-14-28-23(18)22/h4-12,14,20–21H,13,15–16 H2,1–3 H3,(H,29,31)/t20-,21+/m1/s1.

**Appendix 1—chemical structure 11.** *tert*-Butyl (+)-(3*S*,4*R*)−4-(4-iodophenyl)−3-(quinolin-8-ylcarbamoyl)piperidine-1-carboxylate ((+)−7b).

A flame-dried reaction tube was charged with *cis*-3,4-disubstituted piperidine (+)−6b (687 mg, 1.23 mmol, one equiv) and 1,8-diazabicyclo(5.4.0)undec-7-ene (DBU, 550 µL, 3.70 mmol, three equiv). The reaction vessel was sealed with an aluminum cap (with molded butyl/PTFE septa) and purged with argon, then anhydrous toluene (1.20 mL, 1.0 M) was added by syringe. The reaction tube was then placed in a preheated oil bath and stirred at 110°C for 24 hr. The reaction mixture was then allowed to cool to rt and CH$_2$Cl$_2$ (5 mL) and sat. aq. NH$_4$Cl (5 mL) were added. The phases were separated, and the aqueous layer was extracted with CH$_2$Cl$_2$ (3 × 10 mL). The combined organic extracts were dried over Na$_2$SO$_4$ and filtered. The solvent was removed under reduced pressure. The reaction mixture was purified by flash column chromatography (10% acetone/pentane). The product containing fractions were combined and the solvent was removed under reduced pressure. Et$_2$O (10 mL) and pentane (10 mL) were added and the solvent was removed under reduced pressure to afford amide *tert*-butyl (+)-(3*S*,4*R*)−4-(4-iodophenyl)−3-(quinolin-8-ylcarbamoyl) piperidine-1-carboxylate (+)−7b as a white solid (626 mg, 91%, 98.0% *ee*).

$[\alpha]_D^{23}$ + 48.0 (c 1.0, CHCl$_3$).

Characterization data identical to that reported for racemic *trans*-piperidine (±)-S3b (see S24).

HPLC Conditions: Chiralpak IA 3-column, 85:15 *n*-hexane:*i*-PrOH, flow rate: 1 mL·min$^{-1}$, 35°C, UV detection wavelength: 254.1 nm. Retention times: 9.4 min (3*R*,4*S* enantiomer), 13.3 min (3*S*,4*R* enantiomer).

SMILES: O = C([C@@H]1CN(C(OC(C)(C)C)=O)CC[C@H]1C2 = CC = C(I)C = C2)NC3 = C(N = CC = C4)C4 = CC = C3.

InChI = 1S/C26H28IN3O3/c1-26(2,3)33-25(32)30-15-13-20(17-9-11-19(27)12-10-17)21(16-30)24(31)29-22-8-4-6-18-7-5-14-28-23(18)22/h4-12,14,20–21H,13,15–16 H2,1–3 H3,(H,29,31)/t20-,21+/m0/s1.

**Appendix 1—chemical structure 12.** *tert*-Butyl (+)-(*3R,4S*)−4-(4-iodophenyl)−3-(hydroxymethyl) piperidine-1-carboxylate ((+)-S4b) .

A flame-dried reaction tube was charged with amide (–)-S3b (111 mg, 0.20 mmol, one equiv), followed by di-*tert*-butyl dicarbonate (Boc₂O, 175 mg, 0.80 mmol, four equiv) and 4-(dimethylamino) pyridine (DMAP, 4.9 mg, 0.04 mmol, 20 mol %). The reaction vessel was sealed with an aluminum cap (with molded butyl septa) and purged with argon, then anhydrous MeCN (400 µL, 0.5 M) was added by syringe. The mixture was then stirred at 40°C for 22 hr. The reaction mixture was then allowed to cool to rt and sat. aq. NH₄Cl (1 mL) and CH₂Cl₂ (1 mL) were added. The phases were separated, and the aqueous layer was extracted with CH₂Cl₂ (3 × 5 mL). The combined organic extracts were dried over Na₂SO₄ and filtered. The solvent was removed under reduced pressure to afford the crude *N*-Boc protected piperidine derivative.

This crude was solubilized in anhydrous THF (800 µL, 0.2 M) and the resulting solution was added dropwise to a suspension of LiAlH₄ (15.2 mg, 0.40 mmol, two equiv) in anhydrous THF (200 µL, 2.0 M) at 0°C under argon atmosphere. The mixture was then stirred at 20°C for 30 min. The reaction mixture was then quenched by slow addition of sat. aq. NH₄Cl (2 mL) at 0°C and stirred at rt for 30 min. The resulting suspension was filtered through a pad of Celite, eluting with EtOAc (3 × 5 mL). The phases were separated, and the aqueous layer was extracted with EtOAc (3 × 5 mL). The combined organic extracts were dried over Na₂SO₄ and filtered. The solvent was removed under reduced pressure. Purification by flash column chromatography (10% to 15% acetone/pentane) afforded primary alcohol (+)-S4b as a white solid (52.3 mg, 63% over two steps, 98.1% *ee*, containing approx. 10% deiodinated derivative).

$[\alpha]_D^{23}$ + 2.0 (c 1.0, CHCl₃).

R$_f$ 0.24 (15% acetone/pentane); mp = 53–59°C;

$\nu_{max}$ (film)/cm$^{-1}$ 3422 (OH), 2922, 1662 (C = O), 1479, 1424, 1364, 1234, 1163, 1129, 1059, 1006, 816, 764; ¹H NMR (400 MHz, CDCl₃, 298 K) δ 7.66–7.61 (m, 2 hr, HC$_{Ar}$), 6.99–6.93 (m, 2 hr, HC$_{Ar}$), 4.36 (br d, *J* = 13.2 Hz, 1 hr, NC*H*HCHCH₂OH), 4.20 (br s, 1 hr, NC*H*HCH₂), 3.44 (dt, *J* = 11.0, 3.5 Hz, 1 hr, C*H*HOH), 3.26 (dt, *J* = 11.3, 5.8 Hz, 1 hr, CH*H*OH), 2.87–2.63 (m, 2 hr, NCH*H*CHCH₂OH, NCH*H*CH₂), 2.51 (td, *J* = 10.2, 5.2 Hz, 1 hr, CHAr), 1.87–1.72 (m, 2 hr, C*H*CH₂OH, NCH₂C*H*H), 1.71–1.58 (m, 2 hr, NCH₂CH*H*, OH), 1.49 (s, 9 hr, C(CH₃)₃);

¹³C NMR (101 MHz, CDCl₃, 298 K, observed as a mixture of rotamers) δ 154.8 (C = O), 143.5 (C$_{Ar}$ quat), 137.7 (2 × C$_{Ar}$), 129.5 (2 × C$_{Ar}$), 91.7 (IC$_{Ar}$ quat), 79.7 (*C*(CH₃)₃), 63.0 (*C*H₂OH), 46.4 (br m, N*C*H₂CHCH₂OH), 44.4 and 43.5 (N*C*H₂CH₂, *C*HAr, *C*HCH₂OH), 33.8 (NCH₂*C*H₂), 28.5 (C(*C*H₃)₃);

HRMS (ESI⁺) *m/z* Calculated for C₁₇H₂₅NO₃¹²⁷I [M+H] 418.0879; Found 418.0886.

HPLC Conditions: Chiralpak ID 3-column, 90:10 *n*-hexane:*i*-PrOH, flow rate: 1 mL·min⁻¹, 35°C, UV detection wavelength: 230.1 nm. Retention times: 6.7 min (*3R,4S* enantiomer), 7.4 min (*3S,4R* enantiomer).

SMILES: IC1 = CC = C([C@@H]2[C@@H](CO)CN(C(OC(C)(C)C)=O)CC2)C = C1

InChI = 1S/C17H24INO3/c1-17(2,3)22-16(21)19-9-8-15(13(10-19)11−20)12-4-6-14(18)7-5-12/h4-7,13,15,20H,8−11 H2,1−3 H3/t13-,15-/m1/s1

**Appendix 1—chemical structure 13.** *tert*-Butyl (–)-(*3S,4R*)−4-(4-iodophenyl)−3-(hydroxymethyl) piperidine-1-carboxylate ((–)−8b) .

A flame-dried round-bottom flask was charged with amide (+)−7b (558 mg, 1.00 mmol, one equiv), followed by di-*tert*-butyl dicarbonate (Boc$_2$O, 873 mg, 4.00 mmol, four equiv) and 4-(dimethylamino)pyridine (DMAP, 24.4 mg, 0.20 mmol, 20 mol %). The reaction vessel was sealed with an aluminum cap (with molded butyl septa) and purged with argon, then anhydrous MeCN (3.3 mL) and anhydrous CH$_2$Cl$_2$ (0.5 mL) were added by syringe. The mixture (0.3 M) was then stirred at 40°C for 22 hr. The reaction mixture was then allowed to cool to rt and sat. aq. NH$_4$Cl (5 mL) and CH$_2$Cl$_2$ (5 mL) were added. The phases were separated, and the aqueous layer was extracted with CH$_2$Cl$_2$ (3 × 10 mL). The combined organic extracts were dried over Na$_2$SO$_4$ and filtered. The solvent was removed under reduced pressure to afford the crude *N*-Boc protected piperidine derivative.

This crude solubilized in anhydrous THF (3.5 mL, 0.3 M) and the resulting solution was added dropwise to a suspension of LiAlH$_4$ (75.9 mg, 2.00 mmol, two equiv) in anhydrous THF (1.5 mL, 1.0 M) at 0°C under argon atmosphere. The mixture was then stirred at 20°C for 30 min. The reaction mixture was then quenched by slow addition of sat. aq. NH$_4$Cl (5 mL) at 0°C and stirred at rt for 30 min. The resulting suspension was filtered through a pad of Celite, eluting with EtOAc (3 × 10 mL). The phases were separated, and the aqueous layer was extracted with EtOAc (3 × 10 mL). The combined organic extracts were dried over Na$_2$SO$_4$ and filtered. The solvent was removed under reduced pressure. Purification by flash column chromatography (10% to 15% acetone/pentane) afforded primary alcohol (–)−8b as a white solid (315 mg, 68% over two steps, 98.0% *ee*, containing approx. 15% deiodinated derivative).

− $[\alpha]_D^{23}$ 8.0 (*c* 1.0, CHCl$_3$).

Characterization data identical to that reported for enantiomeric alcohol (+)-S4b (see S27).

HPLC Conditions: Chiralpak ID 3-column, 90:10 *n*-hexane:*i*-PrOH, flow rate: 1 mL·min$^{-1}$, 35°C, UV detection wavelength: 230.1 nm. Retention times: 6.7 min (*3R,4S* enantiomer), 7.4 min (*3S,4R* enantiomer).

SMILES: IC1 = CC = C([C@H]2[C@H](CO)CN(C(OC(C)(C)C)=O)CC2)C = C1

InChI = 1S/C17H24INO3/c1-17(2,3)22-16(21)19-9-8-15(13(10-19)11−20)12-4-6-14(18)7-5-12/h4-7,13,15,20H,8−11 H2,1−3 H3/t13-,15-/m0/s1

**Appendix 1—chemical structure 14.** *tert*-Butyl (*3S,4R*)−3-((benzo[d][1,3]dioxol-5-yloxy)methyl)−4-(4-iodophenyl)piperidine-1-carboxylate ((–)−9b) Alcohol (–)−8b (203 mg, 0.49 mmol, one equiv) and triethylamine (96 µL, 0.69 mmol, 1.4 equiv) were added to a flame-dried round-bottom flask, dissolved in anhydrous CH$_2$Cl$_2$ (2.5 mL, 0.2 M) and cooled down to 0°C. Methanesulfonyl chloride (49 µL, 0.64 mmol, 1.3 equiv) was then added by Gilson pipette. After stirring 5 min at 0°C, the reaction mixture was stirred at 25°C for 2 hr, then diluted with CH$_2$Cl$_2$ (5 mL) and sat. aq. NaHCO$_3$ (5 mL). The phases were separated, and the aqueous layer was extracted with CH$_2$Cl$_2$ (3 × 10 mL). The

combined organic extracts were dried over $Na_2SO_4$ and filtered. The solvent was removed under reduced pressure to afford the crude mesylated alcohol derivative.

NaH (60% dispersion in mineral oil, 45.2 mg, 1.10 mmol, 2.2 equiv) was added to a solution of sesamol (135 mg, 0.98 mmol, two equiv) in anhydrous DMF (3.0 mL, 0.3 M) at 0°C. The mixture was then stirred at 25°C for 1 hr. A solution of the crude mesylated alcohol in dry DMF (2.0 mL, 0.2 M) was then added dropwise to this suspension. The resulting mixture was stirred at 90°C for 20 hr. The reaction mixture was quenched by addition of $H_2O$ (5 mL) and aq NaOH 1 N (5 mL) and EtOAc (10 mL) were then added. The phases were separated, and the aqueous layer was extracted with EtOAc (4 × 20 mL). The combined organic extracts were washed with brine (2 × 50 mL), dried over $Na_2SO_4$ and filtered. The solvent was removed under reduced pressure. Purification by flash column chromatography (5% acetone/pentane) afforded piperidine (–)−9b as a white solid (188 mg, 71% over two steps).

– $[\alpha]_D^{23}$ 43.3 (c 1.2, CHCl$_3$).

R$_f$0.15 (5% acetone/pentane); mp = 51–54°C; $\nu_{max}$ (film)/cm$^{-1}$ 2919, 1685 (C = O), 1483, 1424, 1230, 1163, 1129, 1036, 1106, 928, 813, 764; $^1$H NMR (400 MHz, CDCl$_3$, 298 K) δ 7.65–7.59 (m, 2 hr, HC$_{Ar}$), 6.67–6.91 (m, 2 hr, HC$_{Ar}$), 6.64 (d, J = 8.4 Hz, 1 hr, HC$_{Ar}$), 6.36 (d, J = 2.5 Hz, 1 hr, HC$_{Ar}$), 6.14 (dd, J = 8.5, 2.5 Hz, 1 hr, HC$_{Ar}$), 5.89 (s, 2 hr, OCH$_2$O), 4.43 (br s, 1 hr, NCHHCHCH$_2$OAr), 4.25 (br s, 1 hr, NCHHCH$_2$), 3.61 (dd, J = 9.4, 2.9 Hz, 1 hr, CHHOAr), 3.45 (dd, J = 9.4, 6.4 Hz, 1 hr, CHHOAr), 2.91–2.71 (br m, 2 hr, NCHHCHCH$_2$OAr, NCHHCH$_2$), 2.65 (td, J = 11.8, 3.8 Hz, 1 hr, CHAr), 2.08–1.96 (br m, 1 hr, CHCH$_2$OAr), 1.86–1.76 (br m, 1 hr, NCH$_2$CHH), 1.76–1.63 (m, 1 hr, NCH$_2$CHH), 1.50 (s, 9 hr, C(CH$_3$)$_3$);

$^{13}$C NMR (101 MHz, CDCl$_3$, 298 K) δ 154.7 (C = O), 154.2 (OC$_{Ar}$ quat), 148.1 (OC$_{Ar}$ quat), 143.1 (C$_{Ar}$ quat), 141.7 (OC$_{Ar}$ quat), 137.7 (2 × C$_{Ar}$), 129.4 (2 × C$_{Ar}$), 107.8 (C$_{Ar}$), 105.5 (C$_{Ar}$), 101.1 (OCH$_2$O), 98.0 (C$_{Ar}$), 91.8 (IC$_{Ar}$ quat), 79.7 (C(CH$_3$)$_3$), 68.7 (CH$_2$OAr), 47.0 (br m, NCH$_2$CHCH$_2$OAr), 44.3 (NCH$_2$CH$_2$, CHAr), 41.6 (CHCH$_2$OAr), 33.6 (NCH$_2$CH$_2$), 28.4 (C(CH$_3$)$_3$);

HRMS (ESI$^+$) m/z Calculated for C$_{24}$H$_{29}$NO$_5^{127}$I [M+H] 538.1090; Found 538.1104.

SMILES: IC1 = CC = C([C@H]2[C@H](COC3 = CC(OCO4)=C4C = C3)CN(C(OC(C)(C)C)=O)CC2) C = C1.

InChI = 1S/C24H28INO5/c1-24(2,3)31-23(27)26-11-10-20(16-4-6-18(25)7-5-16)17(13-26)14-28-19-8-9-21-22(12-19)30-15-29-21/h4-9,12,17,20H,10–11,13-15H2,1–3 H3/t17-,20-/m0/s1.

**Appendix 1—chemical structure 15.** (3S,4R)−3-((Benzo[d][1,3]dioxol-5-yloxy)methyl)−4-(4-iodophenyl)piperidine-1-ium chloride. (3 • HCl) 4 N HCl in 1,4-dioxane (250 μL, 1.00 mmol, 10 equiv) was added to a solution of N-Boc protected piperidine (–)−9b (56.9 mg, 0.10 mmol) in 1,4-dioxane (250 μL, 0.4 M). At 0°C under air. The solution was stirred at 25°C for 18 hr, then an ice-cold 1:1 mixture of Et$_2$O/pentane (1 mL) was added and formation of a solid precipitate was observed. This was filtered and washed with further ice-cold Et$_2$O/pentane mixture (2 × 5 mL). The solid precipitate was dried under reduced pressure to afford (3S,4R)−3-((benzo[d][1,3]dioxol-5-yloxy)methyl)−4-(4-iodophenyl)piperidine-1-ium chloride 3 • HCl (38.5 mg, 81%) as an off-white solid.

– $[\alpha]_D^{23}$ 86.0 (c 0.9, MeOH). mp = 203–205 °C;

$\nu_{max}$ (film)/cm$^{-1}$ 3321 (NH), 2926, 2807, 1618, 1484, 1185, 1103, 1033, 1003, 932, 846, 813, 787;

$^1$H NMR (400 MHz, CD$_3$OD, 298 K) δ 7.71–7.64 (m, 2 hr, HC$_{Ar}$), 7.11–7.04 (m, 2 hr, HC$_{Ar}$), 6.63 (d, J = 8.5 Hz, 1 hr, HC$_{Ar}$), 6.39 (d, J = 2.5 Hz, 1 hr, HC$_{Ar}$), 6.18 (dd, J = 8.5, 2.5 Hz, 1 hr, HC$_{Ar}$), 5.89–5.82 (m, 2 hr, OCH$_2$O), 3.71–3.62 (m, 2 hr, CHHOAr, NCHHCHCH$_2$OAr), 3.60–3.48 (m, 2 hr, CHHOAr, NCHHCH$_2$), 2.21–2.11 (m, 2 hr, NCHHCHCH$_2$OAr, NCHHCH$_2$), 3.00–2.90 (m, 1 hr, CHAr), 2.49–2.37 (m, 1 hr, CHCH$_2$OAr), 2.09–2.00 (m, 2 hr, NCH$_2$CH$_2$);

$^{13}$C NMR (101 MHz, CD$_3$OD, 298 K) δ 155.2 (OC$_{Ar}$ quat), 149.7 (OC$_{Ar}$ quat), 143.5 (C$_{Ar}$ quat), 143.0 (OC$_{Ar}$ quat), 139.1 (2 × C$_{Ar}$), 130.7 (2 × C$_{Ar}$), 108.8 (C$_{Ar}$), 106.6 (C$_{Ar}$), 102.5 (OCH$_2$O), 98.9 (C$_{Ar}$), 93.1 (IC$_{Ar}$ quat), 68.9 (CH$_2$OAr), 47.7 (NCH$_2$CHCH$_2$OAr), 45.4 (NCH$_2$CH$_2$), 43.0 (CHAr), 40.5 (CHCH$_2$OAr), 31.3 (NCH$_2$CH$_2$);

HRMS (ESI$^+$) *m/z* Calculated for C$_{19}$H$_{21}$NO$_3^{127}$I [M–Cl] 438.0566; Found 438.0571.

SMILES: IC1 = CC = C([C@H]2[C@H](COC3 = CC(OCO4)=C4C = C3)CNCC2)C = C1 .Cl.

InChI = 1S/C19H20INO3.ClH/c20-15-3-1-13(2-4-15)17-7-8-21-10-14(17)11-22-16-5-6-18-19(9-16)24-12-23-18;/h1-6,9,14,17,21H,7–8,10-12H2;1H/t14-,17-;/m0./s1.

