## [Decision Letter]

**Acceptance summary:**

Editors and reviewers were impressed by the scope and detail of your examination of the mode by which the human serotonin transporter recognizes paroxetine, a high-affinity antidepressant. The conclusions reached through this multi-faceted analysis constitute a significant advancement in an exciting field at the interface between membrane physiology and neurobiology.

**Decision letter after peer review:**

Thank you for submitting your article "Chemical and structural investigation of the paroxetine-human serotonin transporter complex" for consideration by *eLife*. Your article has been reviewed by three peer reviewers, and the evaluation has been overseen by Lucy Forrest as the Reviewing Editor and by José Faraldo-Gómez as the Senior Editor. Reviewers #2 and #3 have agreed to reveal their identity: Poul Nissen and Amy Newman, respectively. Based on the reviewers' comments, included below, and on subsequent discussions among reviewers and editors, I am glad to inform you that we believe that your manuscript is potentially suitable for publication in *eLife*, provided you make a number of revisions to address the reviewers' concerns and recommendations, to the extent possible.

We would like to draw your attention to changes in our revision policy that we have made in response to COVID-19 (https://elifesciences.org/articles/57162). Specifically, when editors judge that a submitted work as a whole belongs in *eLife* but that some conclusions require a modest amount of additional new data, as they do with your paper, we are asking that the manuscript be revised to either limit claims to those supported by data in hand, or to explicitly state that the relevant conclusions require additional supporting data. Our hope is that authors will eventually carry out those additional experiments and report on how they affect the relevant conclusions either in a preprint on bioRxiv or medRxiv, or if appropriate, as a Research Advance in *eLife*, either of which would be linked to the original paper.

Reviewer #1:

The manuscript by Gouaux and colleagues contains an analysis of the binding arrangement of the highest-affinity antidepressant, paroxetine, to human serotonin transporter. This ligand has been the subject of a number of previous studies, which suggested that it might adopt one of two alternate arrangements the so-called ABC and ACB poses, in which the two arms of the v-shaped ligand are flipped between the "B" and "C" sub-sites of the central binding site. Prior X-ray crystallographic data however, was limited by the fact that the protein constructs used were of mutants than were modified in key locations.

The combination of cryo-EM, x-ray structural analysis with synthesis of brominated and iodinated compounds, binding (*Citalopram competition) and 5HT uptake assays is impressive. Anomalous diffraction identifies some signal for the halogenated group in the location consistent with the ABC pose; however, computational analyses by Abagayan suggested the bromination itself tends to favor this pose. The most compelling data therefore comes from the ~3.3Å cryo-EM structures of the wild-type protein bound to paroxetine, which indicate differences in the correlation coefficient for fitting the ligand in the two orientations. Unfortunately, at this resolution, I believe this data cannot be used to rule out the possibility that both orientations are occupied, especially if ACB has a lower occupancy.

I conclude that these data are further suggestive of the ABC pose being the primary arrangement, but do not categorically rule out that paroxetine also adopts the ACB pose.

Major comments:

1) The anomalous diffraction densities are poorly defined (at 4 sigma), albeit appearing in the general region expected for the position of paroxetine that the authors identify. I am also concerned by the large fraction of residues that are defined as containing a poor fit to the EM map (all-atom inclusion <40%). Is this some kind of model bias, because of using the previously obtained X-ray structures as starting points for refinement? (MR using 6AWN = S439T/3.6Å).

2) Discussion paragraph four starting: "Paroxetine is stabilized...". These arguments read as self-fulfilling. Close contact in a structure does not mean that all interactions are stabilizing; it neglects entropic contributions and the fact that some contacts may be disfavored in order to optimize the overall free energy. Perhaps it would be useful to separate the descriptions of the interactions with the discussion of the nature of the ABC and ACB orientations. For example, the arguments regarding the ABC and ACB orientations seem to relate to the presence of aromatic groups and hydrogen-bonding of the benzodioxol and fluorophenyl moieties; but wouldn't both types of interactions be present also in the ACB mode? Similarly, why are differences in electronegativity between C-F, C-Br and C-I more relevant for subsite C rather than subsite B?

3) Remarkably, and somewhat disturbingly, in the new cryo-EM structures the tail of DDM protrudes between helices TM11/10 and TM12, which are in turn more separated. How is it possible that the same tail is not observed/modelled in any of the previous X-ray structures, even though the maltose group is modeled in the same position in e.g. 6AWN? This discrepancy should be described and mentioned in the Results when comparing the cryo-EM and x-ray structures.

Reviewer #2:

Coleman et al. present a study of paroxetine (pax) binding investigated with different methods (single-particle cryo-EM, X-ray crystallography, binding studies) using five different constructs of SERT – i) wild-type with C-term GFP, ii) ts2 active with Ile291Ala, Thr439Ser and C-term GFP, iii) same ts2 active with in addition Asn177 mutations, iv) inactive ts2 with Tyr110Ala, Ile291Ala and v) ΔN72, ΔC13 SERT plus an antibody 8B6. They investigate the binding of Br- and I-pax derivatives (Br and I substituting F in the fluorophenyl-group of pax), the synthesis of which is also described here. These derivative compounds bind and inhibit SERT, although with a somewhat reduced affinity (going from low to high nM range). Crystal structures and cryo-EM structures of different construct and complexes are overall very consistent on how pax binds to SERT.

The study is motivated by ambiguities in the literature on how pax binds, represented by ABC or ACB poses to subsites A,B, and C at the central site of SERT. Earlier structural studies also from the Gouaux lab as well as the new studies presented here show the ABC pose, while docking studies and MD simulation from the Shi group indicate a more dynamic binding mode featuring both ABC and ACB poses that through proposed entropy effects for the pseudosymmetry of pax would explain a high affinity. The study presented in this manuscript is important and relevant to a large readership, and to applied uses of the structural information for further drug development.

Subsite A of SERT binds an amine functionality, here the piperidine ring of pax. The B asubsite attracts bulky groups with electronegative substituents such as halogenated aromatic groups, catecholes and the benzodioxol group of pax. Site C occupies other groups, in pax a halogenated aromatic group (fluoro-phenyl). Pax thus has two aromatic/heterocylic groups with electronegative substituents – the fluoro-phenyl group and the benzodioxol, and according to the structural studies the benzodioxol "wins" the B-subsite. Studies of the ABC vs. ACB pose dynamics through derivative compounds should perturb the balance of how the two subsites B and C are occupied – e.g. by making the benzodioxol group bind weaker to the B-subsite or the fluorophenyl group potentially binding stronger to it. The study design however decreases the electronegativity of the fluorophenyl group by substituting F for Br or I – this will presumably make it bind weaker to both subsites B and C. The structures, therefore, and perhaps not surprisingly, confirm the ABC pose found in earlier structural studies. Studies of Asn177 mutants in the ts2 active background show similar consistency to this binding mode and the study leaves little doubt that the ABC pose may be a dominating pose.

Another question has been raised on the use of the thermostabilizing mutation Thr439Ser, which according to the Shi group obscures the dynamic nature of pax binding by ABC/ACB poses and again the authors find no major differences based on this mutant form.

Major points

1) It is rather difficult to keep track of the SERT constructs used throughout the report – please include/add a table and perhaps also a figure that the reader can refer to and be careful that the text is always clear on which construct is now described/discussed

2) It may be argued that the study design does not really challenge the ambiguity of earlier studies. It would be critical to include other pax derivatives, eg. some that reduce the electronegativity of heteroatoms of the benzodioxol group, which will then likely loose affinity to the subset B, or that increase the B-subsite affinity of the fluoro-phenyl group (e.g. as a 3,4-difluorophenyl).

Alternatively, the manuscript must be rewritten into a report that confirms the ABC pose of pax observed under the given set of conditions – that rather than opposing the ACB pose model.

3) The cryo-EM structures represent only a small fraction of total particles in the cryo-EM data sets (about 10%). What else is observed from the remaining 90% of the datasets? Could other classes of the cryo-EM study of ΔN72, ΔC13 represent more flexible structures with ACB poses of pax?

4) The anomalous difference map analysis is scarcely described and therefore difficult to review. Anomalous difference maps must be carefully assessed to obtain a maximal signal-noise ratio, which on the other hand is crucial for any statements to be made on the absence of minor sites (which will disappear in the noise of a suboptimal fourier analysis). The S/N ratio might for example (conveniently) be assessed by a comparison of the highest positive (signal) compared to the highest negative peaks (noise) in a map. Table 4 indicate that the resolution of the data has been stretched to a maximum (and remains low resolution), and that atomic models have been refined against these very low resolution data for Br-pax and I-pax (compared to the earlier ts3-pax data at 3.14 Å resolution in a similar crystal form). Reading from the manuscript, model phases have been derived then from these low-resolution refined structures and used for the anomalous difference Fourier map. This is not necessarily the way to get the highest S/N ratio, where higher quality model phases from a higher-resolution isomorphous structure can also be applied, perhaps combined with an appropriate blurring of the phases at higher resolution by a negative temperature factor approach – this must be tested. Furthermore, the exact choice of resolution of the input anomalous data must be established by trial-and-error searching again for the maximal S/N ratio of the anomalous difference map, and also other parameters (such as a maximum threshold allowed for F+/-F- differences). The maps must be shown for both subsites B and C

5) All paroxetine crystal forms seem more or less isomorphous – an isomorphous difference Fourier map analysis (Fo-Fo maps) would be very useful, and will often show minor sites well. Any relevant isomorphous pair of paroxetine crystals (Br-pax/pax; Br-pax/I-pax, I-pax/pax) should be analysed. Again, the model phases used for the Fourier analysis should be selected carefully and probably come from the higher resolution "native pax structure". Similar to the anomalous difference maps, the isomorphous difference maps must be shown for subsites B and C

Reviewer #3:

The authors combined a novel chemical synthesis of previously described Br-paroxetine and its I-analogue, cryo-EM, and crystallography to further probe the SSRI paroxetine binding pose at the central site of SERT. While the experimental design and the data collection are solid, the deduction toward the conclusion does not entirely resolve the controversy originally described in the Introduction for paroxetine itself.

Substantive but addressable concerns:

1) Overall the cryo-EM structures are of low resolution and the authors could not "identify features associated with the scattering of bromine and iodine". The crystal structures in the "ts2-inactive" background are of even lower resolutions (4.69 and 6.12Å), though there were "clear density for Br- and I- atoms of the paroxetine derivatives in subsite C". Thus, given that the CC of ACB pose, 0.70, is not unacceptably low, there is no definitive evidence presented to specifically argue for the ABC and against the ACB pose. In addition, as -F to -Br and -I may significantly change the properties of the compound (see Abramyan, et al., 2018), it is not safe to assume that Br- or I-paroxetine are in the same pose as paroxetine. See https://www.ncbi.nlm.nih.gov/pubmed/27982595. Indeed, it is difficult for the cryo-EM and X-ray crystallography work to consider dynamics, thus the authors should not completely exclude the possibility of an ACB pose (see Discussion).

2) Importantly, there appear to be some misinterpretations of the studies described in Abramyan et al., 2018 and Slack et al., 2019, that must be addressed:

• "However, recent mutagenesis, molecular dynamics, and binding studies with paroxetine analogues suggest that paroxetine may occupy two distinct poses in which the benzodioxol and fluorophenyl groups reside in subsite B or C, depending on the rotameric position of Phe341 and the presence of the thermostabilizing mutation Thr439Ser18,20 (ACB pose, Figure 1C)." The main points in Abramyan et al., 2018 and Slack et al., 2019, were that the favored entropy component may play a significant role in paroxetine's high affinity, which include the dynamics in the S1 site (e.g., the dynamics of Phe341 in the presence of paroxetine but not Br-paroxetine) and potential pose ambiguity. The mutations may disrupt such dynamics but not necessarily associated with a specific pose.

• Regarding the synthesis of Br- and I-paroxetine analogues, this alternative approach is indeed new, however the authors should acknowledge that Br-paroxetine was previously synthesized and fully characterized in Slack et al., 2019.

• "In this study, the authors hypothesized that the difference could be because of the crystallization conditions and thermostabilizing mutations." It is not clear what "this study" is referring to. Davis et al., 2016, did not discuss the crystal structures, while Abramyan et al did not hypothesize that WT should be in the ACB pose (see above).

• Importantly, the dominant occupation of Br-paroxetine in the ABC pose is consistent with the findings in Slack et al., 2019 and should be acknowledged, as the studies in the present report for this compound were why it was synthesized in the first place. Based on the molecular dynamics studies in Abramyan et al, the studies in the present report neither support nor refute that paroxetine may bind in either pose in the native hSERT.

---

## [Author Response]

Reviewer #1:The manuscript by Gouaux and colleagues contains an analysis of the binding arrangement of the highest-affinity antidepressant, paroxetine, to human serotonin transporter. This ligand has been the subject of a number of previous studies, which suggested that it might adopt one of two alternate arrangements the so-called ABC and ACB poses, in which the two arms of the v-shaped ligand are flipped between the "B" and "C" sub-sites of the central binding site. Prior X-ray crystallographic data however, was limited by the fact that the protein constructs used were of mutants than were modified in key locations.The combination of cryo-EM, x-ray structural analysis with synthesis of brominated and iodinated compounds, binding (*Citalopram competition) and 5HT uptake assays is impressive. Anomalous diffraction identifies some signal for the halogenated group in the location consistent with the ABC pose; however, computational analyses by Abagayan suggested the bromination itself tends to favor this pose.

We have added a sentence about the computational analysis of Abagayan, see “These studies also suggested that bromination of paroxetine and certain mutations near the central site, such as Ala169Asp, favored ABC pose.” While we agree with the reviewer that a stronger anomalous difference peak would have been helpful toward understanding if the ACB pose is present at low occupancy, the observed density in subsite C for the halogens is substantial. Based on comments from reviewer #2 we have increased the signal-to-noise ratio of the anomalous maps and also have provided isomorphous difference maps which further support the ABC pose as being the favored pose for paroxetine and the analogues investigated in this work.

The most compelling data therefore comes from the ~3.3Å cryo-EM structures of the wild-type protein bound to paroxetine, which indicate differences in the correlation coefficient for fitting the ligand in the two orientations. Unfortunately, at this resolution, I believe this data cannot be used to rule out the possibility that both orientations are occupied, especially if ACB has a lower occupancy.

We have not ruled out the ACB pose and have revised the manuscript to make it clear that the ACB pose could be present at an occupancy that is not discernible at the present resolution.

I conclude that these data are further suggestive of the ABC pose being the primary arrangement, but do not categorically rule out that paroxetine also adopts the ACB pose.

As mentioned above, we agreed with the reviewer that our data does not rule out the ACB pose but rather supports the conclusion that the ABC pose is the dominant pose.

Major comments:1) The anomalous diffraction densities are poorly defined (at 4 sigma), albeit appearing in the general region expected for the position of paroxetine that the authors identify. I am also concerned by the large fraction of residues that are defined as containing a poor fit to the EM map (all-atom inclusion <40%). Is this some kind of model bias, because of using the previously obtained X-ray structures as starting points for refinement? (MR using 6AWN = S439T/3.6Å).

Considering the resolution of Br-paroxetine and I-paroxetine structures, the fact that the highest anomalous difference peak goes beyond 4σ is meaningful. The Br- peak is now at 6.0σ and I- peak is at 4.5σ. We would like to draw the reviewer’s attention to Figure 4—figure supplement 5, which shows that most residues fit the density well. The low all-atom inclusion score is because of unmodeled micelle density and the unmodeled constant domain of Fab.

2) Discussion paragraph four starting: "Paroxetine is stabilized…". These arguments read as self-fulfilling. Close contact in a structure does not mean that all interactions are stabilizing; it neglects entropic contributions and the fact that some contacts may be disfavored in order to optimize the overall free energy. Perhaps it would be useful to separate the descriptions of the interactions with the discussion of the nature of the ABC and ACB orientations. For example, the arguments regarding the ABC and ACB orientations seem to relate to the presence of aromatic groups and hydrogen-bonding of the benzodioxol and fluorophenyl moieties; but wouldn't both types of interactions be present also in the ACB mode? Similarly, why are differences in electronegativity between C-F, C-Br and C-I more relevant for subsite C rather than subsite B?

We acknowledge that these interactions, in a broad sense, could be possible in both poses. However, we observe that there also would be significant changes in the distances of many interactions upon modeling paroxetine in the ACB vs. the ABC pose. The differences in electronegativity and carbon-halogen bond length are relevant to understand the differences in affinities of paroxetine derivatives irrespective of the pose. The Discussion section has been revised to reflect these views.

3) Remarkably, and somewhat disturbingly, in the new cryo-EM structures the tail of DDM protrudes between helices TM11/10 and TM12, which are in turn more separated. How is it possible that the same tail is not observed/modelled in any of the previous X-ray structures, even though the maltose group is modeled in the same position in e.g. 6AWN? This discrepancy should be described and mentioned in the Results when comparing the cryo-EM and x-ray structures.

The head group positioning in the x-ray structures suggests that the tail group of detergent would have extended into the space between TM10, 11, and 12. However, the tail was not modeled because of lack of density in the x-ray structure. Differences in the way maps are obtained and in the experimental conditions (i.e. DDM vs. OM) in both experiments make it possible that the same feature, such as a detergent tail, is visible in the EM map but not in the xray map. We have added a statement describing the differences between likely detergent density in the cryo-EM and x-ray structures: “In the cryo-EM maps, the maltose headgroup of a DDM molecule could also be visualized in the allosteric site with the detergent tail inserted between TMs 10, 11, and 12. In contrast, in the x-ray maps only the head group of the octyl-maltoside detergent could be modeled due to the weak density of the hydrocarbon chain.”

We also note that the conformations of TMs 10, 11, and 12 are similar to their respective conformations in other outward-open SERT structures solved by cryo-EM. Please see Figure 5.

Reviewer #2:[…]Major points1) It is rather difficult to keep track of the SERT constructs used throughout the report – please include/add a table and perhaps also a figure that the reader can refer to and be careful that the text is always clear on which construct is now described/discussed.

Table 1 has been added in the revised version of the manuscript describing the SERT variants used in this study.

2) It may be argued that the study design does not really challenge the ambiguity of earlier studies. It would be critical to include other pax derivatives, eg. some that reduce the electronegativity of heteroatoms of the benzodioxol group, which will then likely loose affinity to the subset B, or that increase the B-subsite affinity of the fluoro-phenyl group (e.g. as a 3,4-difluorophenyl).Alternatively, the manuscript must be rewritten into a report that confirms the ABC pose of pax observed under the given set of conditions – that rather than opposing the ACB pose model.

We appreciate reviewer’s insights on the manuscript. However, structural studies involving other derivatives of paroxetine are beyond the scope of this manuscript. The data that we have presented in the manuscript demonstrates that under the examined conditions, the ABC pose of paroxetine is favored over the ACB pose. However, we agree with reviewer that we cannot exclude the possibility of ACB pose. The manuscript has been rephrased accordingly.

3) The cryo-EM structures represent only a small fraction of total particles in the cryo-EM data sets (about 10%). What else is observed from the remaining 90% of the datasets? Could other classes of the cryo-EM study of ΔN72, ΔC13 represent more flexible structures with ACB poses of pax?

The particle picking was carried out using two different methods – DoG-picker and blobbased picking in cryoSPARC, which both resulted in about 2 million picks. Most of these picks are not SERT particles. When we performed 2D classification on the picks that were discarded, we found that they did not have any features of SERT-Fab complexes. We sorted the picks by different methods using 3D and 2D classification and then combined them for refinement. About 120K particles were removed using 3D classification for the paroxetine data set because reconstructions derived from these particles were poorly resolved and it was not possible to observe a density feature for paroxetine. It is possible that these particles are more flexible with an ACB pose, though it is also likely that they did not produce higher resolution reconstructions due to other factors such as denaturation at the air-water interface, thick or poor quality ice, or difficulties in determination of accurate particles parameters. The final particle set has 420K particles for paroxetine, which is about 20% of the total picks, a similar percentage of the total picks across all three data sets.

For the Br and I-paroxetine data sets, all the particles which resembled SERT-Fab from 2D were used in refinement. Hence, we believe that the final dataset is a good representation of the major conformation seen in the dataset. We do agree that in the classes that were excluded during 2D and 3D classification steps that it is still possible that some of the discarded particles were more flexible with an ACB pose of paroxetine at the central site. However, the excluded classes were poorly resolved, hence making any high-resolution inferences from them ambiguous. Owing to this possibility we rephrased certain portions of the manuscript to allow for the possibility of ACB pose.

“We observed that in the ACB pose, paroxetine could be positioned with a CC of 0.70 compared with 0.84 for the ABC pose suggesting that while ABC pose is clearly preferred under the conditions we tested, the possibility of an ACB pose cannot be excluded (Figure 4—figure supplementary 5A,B).”

See also:

“However, the data presented in the manuscript does not completely exclude the possibility of an ACB pose at the central site.”

4) The anomalous difference map analysis is scarcely described and therefore difficult to review. Anomalous difference maps must be carefully assessed to obtain a maximal signal-noise ratio, which on the other hand is crucial for any statements to be made on the absence of minor sites (which will disappear in the noise of a suboptimal fourier analysis). The S/N ratio might for example (conveniently) be assessed by a comparison of the highest positive (signal) compared to the highest negative peaks (noise) in a map. Table 4 indicate that the resolution of the data has been stretched to a maximum (and remains low resolution), and that atomic models have been refined against these very low resolution data for Br-pax and I-pax (compared to the earlier ts3-pax data at 3.14 Å resolution in a similar crystal form). Reading from the manuscript, model phases have been derived then from these low-resolution refined structures and used for the anomalous difference Fourier map. This is not necessarily the way to get the highest S/N ratio, where higher quality model phases from a higher-resolution isomorphous structure can also be applied, perhaps combined with an appropriate blurring of the phases at higher resolution by a negative temperature factor approach – this must be tested. Furthermore, the exact choice of resolution of the input anomalous data must be established by trial-and-error searching again for the maximal S/N ratio of the anomalous difference map, and also other parameters (such as a maximum threshold allowed for F+/-F- differences). The maps must be shown for both subsites B and C

We thank the reviewer for these suggestions. Although we had already performed many of these analyses to improve the S/N ratio of the anomalous maps in the initial submission, we had not described what we had done in sufficient detail. Furthermore, we had not tried to blur the phases at higher resolution, this was a helpful suggestion and improved the quality of the Brparoxetine map substantially (maximum peak intensity 5.5 vs. 6.0σ). We have also written a paragraph in the Materials and methods section to describe what we have done in more detail:

“Anomalous difference maps

X-ray data sets were processed with XDS; Friedel pairs were allowed to have different intensities. Molecular replacement was performed with coordinates from the previously determined ts2-inactive SERT-paroxetine structure (Protein Data Bank (PDB) code: 6AWN) using PHASER. B-factors were refined using PHENIX followed by generating anomalous difference maps using the phases derived from the higher resolution structures. To maximize the signal-to-noise ratio of the Br-paroxetine anomalous difference density, the high-resolution phases were blurred with a B-factor of 500 with a high-resolution cutoff of 5.5 Å. Using these optimized parameters for the Fourier analysis of the Br-paroxetine diffraction data, we obtained an anomalous map with the largest difference peak being present at 6.0σ and the noise level estimated at ~2.5σ. To maximize the signal-noise-ratio of the I-paroxetine anomalous difference density, a high-resolution and low-resolution cutoff of 6.3 and 30 Å was applied during the generation of the anomalous maps. Using these optimized parameters for the Fourier analysis of the I-paroxetine diffraction data, we obtained an anomalous map with the largest difference peak being present at 4.5σ and the noise level estimated at ~2.5σ.”

5) All paroxetine crystal forms seem more or less isomorphous – an isomorphous difference Fourier map analysis (Fo-Fo maps) would be very useful, and will often show minor sites well. Any relevant isomorphous pair of paroxetine crystals (Br-pax/pax; Br-pax/I-pax, I-pax/pax) should be analysed. Again, the model phases used for the Fourier analysis should be selected carefully and probably come from the higher resolution "native pax structure". Similar to the anomalous difference maps, the isomorphous difference maps must be shown for subsites B and C

We appreciate this constructive suggestion. The paroxetine crystal forms are indeed fairly isomorphous with <1% difference in unit cell dimensions. When we analyzed the suggested Fo(paroxetine)-Fo(Br-paroxetine) and Fo(paroxetine)-Fo(I-paroxetine) maps, we found that they did contain density features of the halogens. The Fo(Br-paroxetine)-Fo(I-paroxetine) maps, did not have meaningful features, perhaps because of the low resolution of both datasets. Like the anomalous difference maps, we have also shown both subsite B and C in Figure 4—figure supplement 6 and have displayed the difference density with a radius covering subsite B and C. We added the following text:

“Next, we calculated isomorphous difference maps (F_o_-F_o_) using the ts2-inactive paroxetine dataset (PDB: 6AWN) and either the Br-paroxetine or I-paroxetine datasets. The F_o_(paroxetine)F_o_(Br-paroxetine) map also revealed a difference peak in subsite C near the halogenated groups while no significant peaks were detected in subsite B (Figure 4—figure supplement 6A). Similarly, the F_o_(paroxetine)-F_o_(I-paroxetine) map also contained a difference peak which overlapped with the position of the halogen (Figure 4—figure supplement 6B) while the F_o_(Brparoxetine)-F_o_(I-paroxetine) difference map did not contain any interpretable features, likely due to the low resolution of both datasets (Figure 4—figure supplement 6C).”

We have added a section in the Materials and methods to describe in detail how this analysis was performed,

**“**Fo-Fo isomorphous difference maps

Isomorphous difference (F_o_-F_o_) maps were calculated in PHENIX by analyzing isomorphous pairs of crystals. Difference maps were calculated using the previously determined ts2-inactive SERT-paroxetine dataset and PDB (6AWN) for phasing. High- and low-resolution cutoffs of 6.0 and 30.0 Å were applied for the F_o_(paroxetine)- F_o_(Br-paroxetine) map and cutoffs of 6.3 and 30.0 Å were used for the F_o_(paroxetine)- F_o_(I-paroxetine) and F_o_(Br-paroxetine)- F_o_(Iparoxetine) maps.”

Reviewer #3:The authors combined a novel chemical synthesis of previously described Br-paroxetine and its I-analogue, cryo-EM, and crystallography to further probe the SSRI paroxetine binding pose at the central site of SERT. While the experimental design and the data collection are solid, the deduction toward the conclusion does not entirely resolve the controversy originally described in the Introduction for paroxetine itself.Substantive but addressable concerns:1) Overall the cryo-EM structures are of low resolution and the authors could not "identify features associated with the scattering of bromine and iodine". The crystal structures in the "ts2-inactive" background are of even lower resolutions (4.69 and 6.12Å), though there were "clear density for Br- and I- atoms of the paroxetine derivatives in subsite C". Thus, given that the CC of ACB pose, 0.70, is not unacceptably low, there is no definitive evidence presented to specifically argue for the ABC and against the ACB pose. In addition, as -F to -Br and -I may significantly change the properties of the compound (see Abramyan, et al., 2018), it is not safe to assume that Br- or I-paroxetine are in the same pose as paroxetine. See https://www.ncbi.nlm.nih.gov/pubmed/27982595. Indeed, it is difficult for the cryo-EM and X-ray crystallography work to consider dynamics, thus the authors should not completely exclude the possibility of an ACB pose (see Discussion).

We appreciate this comment and given the moderate resolution of the x-ray data, the reviewer is correct that we would not expect to see the ACB pose if it is present at low occupancy. The CC of the ACB pose is 0.7 compared to 0.84 of the ABC pose and thus we agree that we cannot exclude the possibility of the ACB pose. We note that both via x-ray crystallography and cryo-EM that all three variants of paroxetine occupy a pose that could be best described as ABC pose, under the conditions of structure determination. Nonetheless, we agree that it is not possible to exclude ACB pose based on the data that we presented in the manuscript. The Discussion section has been edited to reflect these changes.

2) Importantly, there appear to be some misinterpretations of the studies described in Abramyan et al., 2018 and Slack et al., 2019, that must be addressed:• "However, recent mutagenesis, molecular dynamics, and binding studies with paroxetine analogues suggest that paroxetine may occupy two distinct poses in which the benzodioxol and fluorophenyl groups reside in subsite B or C, depending on the rotameric position of Phe341 and the presence of the thermostabilizing mutation Thr439Ser18,20 (ACB pose, Figure 1C)." The main points in Abramyan et al., 2018 and Slack et al., 2019, were that the favored entropy component may play a significant role in paroxetine's high affinity, which include the dynamics in the S1 site (e.g., the dynamics of Phe341 in the presence of paroxetine but not Br-paroxetine) and potential pose ambiguity. The mutations may disrupt such dynamics but not necessarily associated with a specific pose.

This statement in the Introduction, has now been edited to read as follows “However, recent mutagenesis, molecular dynamics, and binding studies with paroxetine analogues suggest that paroxetine might either occupy the ABC pose as observed in the crystal structure, or an ACB pose where the benzodioxol and fluorophenyl groups occupy subsite C and B of the central site respectively (Abramyan et al., 2019; Slack et al., 2019) (Figure 1C).”

• Regarding the synthesis of Br- and I-paroxetine analogues, this alternative approach is indeed new, however the authors should acknowledge that Br-paroxetine was previously synthesized and fully characterized in Slack et al., 2019.

We have now acknowledged the previous reports mentioning synthesis and complete characterization of Br-paroxetine in the Results section. We added a statement that says “In contrast, common methods for (–)-paroxetine synthesis can require the aromatic substituent to be introduced before stereoselective steps or ring construction, reducing flexibility of the process (Slack et al., 2019; Johnson et al., 2001; Hughes, Kimura and Buchwald, 2003; Brandau et al., 2006; Krautwalk et al., 2014; Wang et al., 2015; Kubota et al., 2016; Amat et al., 2000), nevertheless, Br-paroxetine and other analogues of paroxetine have been previously synthesized by these methods and binding to SERT has been extensively studied (Slack et al., 2019; Brandau et al., 2006).”

• "In this study, the authors hypothesized that the difference could be because of the crystallization conditions and thermostabilizing mutations." It is not clear what "this study" is referring to. Davis et al., 2016, did not discuss the crystal structures, while Abramyan et al. did not hypothesize that WT should be in the ACB pose (see above).

We have edited the last paragraph to read:

“However, computational docking experiments using wild-type SERT predicted that the position of benzodioxol and fluorophenyl groups of paroxetine are “flipped”, with paroxetine occupying an ACB pose (David et al., 2016) (Figure 1C). Subsequent studies involving wild-type and mutant SERT variants that include modelling, mutagenesis, and Br-paroxetine docking experiments suggested that paroxetine could bind in both ABC and ACB poses. These studies also suggested that bromination of paroxetine and certain mutations near the central site, such as A169D, favored ABC pose (Abramyan et al., 2018; Slack et al., 2019). Hence, the authors in these studies hypothesized that the ABC pose observed in the crystal structure could be because of the crystallization conditions and thermostabilizing mutations.”

• Importantly, the dominant occupation of Br-paroxetine in the ABC pose is consistent with the findings in Slack et al., 2019 and should be acknowledged, as the studies in the present report for this compound were why it was synthesized in the first place. Based on the molecular dynamics studies in Abramyan et al., the studies in the present report neither support nor refute that paroxetine may bind in either pose in the native hSERT.

In multiple SERT structures, solved using both x-ray crystallography and cryo-EM, all three (F-, Br-, and I-) variants of paroxetine at the central site could be interpreted to be in an ABC pose. Based on these observations, we believe that bromination alone is not the reason for the ABC pose. However, we have now edited the text in Discussion and Results to indicate that while the ABC pose is favored in the conditions that we solved the structures, the possibility of the ACB pose cannot be excluded.